# Computational design of dynamic receptor–peptide signaling complexes applied to chemotaxis

Robert E. Jefferson [1,2], Aurélien Oggier[1,2], Andreas Füglistaler [1,2], Nicolas Camviel[2,3], Mahdi Hijazi [1,2], Ana Rico Villarreal[1,2], Caroline Arber [2,3] & Patrick Barth [1,2] ✉

Engineering protein biosensors that sensitively respond to specific biomolecules by triggering precise cellular responses is a major goal of diagnostics and synthetic cell biology. Previous biosensor designs have largely relied on binding structurally well-defined molecules. In contrast, approaches that couple the sensing of flexible compounds to intended cellular responses would greatly expand potential biosensor applications. Here, to address these challenges, we develop a computational strategy for designing signaling complexes between conformationally dynamic proteins and peptides. To demonstrate the power of the approach, we create ultrasensitive chemotactic receptor–peptide pairs capable of eliciting potent signaling responses and strong chemotaxis in primary human T cells. Unlike traditional approaches that engineer static binding complexes, our dynamic structure design strategy optimizes contacts with multiple binding and allosteric sites accessible through dynamic conformational ensembles to achieve strongly enhanced signaling efficacy and potency. Our study suggests that a conformationally adaptable binding interface coupled to a robust allosteric transmission region is a key evolutionary determinant of peptidergic GPCR signaling systems. The approach lays a foundation for designing peptide-sensing receptors and signaling peptide ligands for basic and therapeutic applications.

Designing biosensors with arbitrary input and output behaviors is a grand challenge of synthetic biology. Current approaches focus on engineering binding to structurally well-defined protein[1] and small-molecule chemical cues[2], and couple molecular recognition to synthetic optical reporters that are built-in modular biosensor scaffolds. While this strategy provides elegant solutions to the design of in vitro diagnostics, applications for in vivo detection and synthetic cell biology rely on coupling the molecular sensor to the precise activation and orchestration of complex intracellular signaling functions that often cannot be recapitulated de novo. Harnessing synthetic sensing to fine-

tuned native signaling functions in a biosensor scaffold is limited by our poor mechanistic understanding of allosteric signal transduction and lack of techniques to rationally engineer these properties.

Computational approaches for the design of protein-protein recognition have produced a wide array of therapeutic proteins including potent inhibitors and vaccines mostly through the optimization of binding interactions between static protein surfaces[2–4]. However, several classes of proteins including signaling receptors and peptides display high levels of conformational plasticity and binding of these molecules often involves large structural rearrangements

[1]Interfaculty Institute of Bioengineering, École Polytechnique Fédérale de Lausanne, Lausanne CH-1015, Switzerland. [2]Ludwig Institute for Cancer Research Lausanne, Lausanne, Switzerland. [3]Department of Oncology UNIL-CHUV, University Hospital Lausanne (CHUV), University of Lausanne (UNIL), Lausanne, Switzerland. ✉e-mail: patrick.barth@epfl.ch

through conformational selection and mutual induced fit[5–8]. The rational design of dynamic binding complexes remains particularly challenging and has not been reported to date.

Peptides mediate close to 40% of cell signaling functions through ubiquitous interactions with membrane receptors and soluble proteins[9,10]. Unbound peptide ligands are often partially disordered in solution, which challenges structure determination and the computational sampling of the vast space of peptide conformations. In contrast to rigid protein binders and small-molecule ligands, structural information on peptide binding is scarce and limits supervised training and validation of deep-learning[11–13] and physics-based[14] protein–peptide complex structure prediction approaches. Consequently, the mechanistic underpinnings of peptide-mediated functions remain also poorly understood.

A recent comparative genomics study of peptidergic GPCRs revealed important features of the peptide-GPCR network[15]. Peptide-binding GPCRs typically involve larger binding cavities and ligand contact areas than receptors binding to small molecules. The peptidergic signaling network is often characterized by GPCRs sensing an array of peptide ligands and peptides capable of activating several receptors, which complicates the prediction of binding and signaling determinants. The specific receptor–peptide modeling and engineering problem is further complicated by the high flexibility of both receptor and peptide ligand, which often mutually adopt a new conformation through induced fit to reach the active state and initiate signal transduction[16].

In this study, we first develop a computational strategy for modeling the binding between flexible peptides and structurally uncharacterized proteins and designing signaling membrane receptors with high binding sensitivity to peptide ligands. To validate the approach, we create chemokine receptor–peptide pairs that elicit potent intracellular signaling in human cells and chemotactic responses in primary T cells. Lastly we carry out molecular dynamics simulations on the complexes and uncover mechanistic determinants of GPCR–peptide recognition and signaling.

## Results

### Overall rationale and goal of the study

In the long run, we aim to design custom-built modular biosensors that can link binding of a flexible peptide input signal to fine-tuned and complex cellular responses through genetically encoded single-receptor domains. We define this designed class of biosensors as CAPSens, which stands for Conformationally Adaptive Peptide Bio-Sensors. Such an approach would enable the reprogramming of cellular functions upon a wide range of environmental cues and would impact cellular therapies that rely on cell trafficking, including cancer immunotherapies.

Toward that goal, we developed a method that can build flexible receptor–peptide conformational ensembles and model peptide-mediated receptor signaling pathways. Unlike previous work that mainly optimizes binding and models receptors as rigid target structures[17], this approach enables the modeling of signaling active states and the design of dynamic complexes with altered binding contacts and allosteric networks enhancing both recognition sensitivity and signaling response (Fig. 1a, b).

To demonstrate this strategy, we targeted the chemokine receptor CXCR4–CXCL12 peptide signaling axis. We selected that signaling complex because CXCR4, upon sensing its native ligand CXCL12, regulates important physiological functions, including cell chemotaxis (i.e., cell migration along a gradient of CXCL12), but remains structurally uncharacterized in the active signaling state. Using the approach, we modeled and designed CXCR4 variants with high binding sensitivity to the native CXCL12 and also created receptor–peptide binding pairs that triggered potent signaling and cell migration (Fig. 1b).

## Computational modeling and design framework of GPCR–peptide signaling complexes

Despite tremendous progress in protein structure determination, experimental structures of signaling receptor–peptide complexes remain scarce. In absence of structures of the interacting partners, the design of binding complexes necessitates a method that both models the conformations of the bound molecules and engineers functional binding interactions. Molecular recognition between flexible peptide and signaling receptors likely involves significant structural rearrangements of both molecules through conformational selection (i.e., selection from an ensemble of unbound conformations) and induced fit (i.e., conformational changes occurring upon binding) effects. Therefore, we first reasoned that an effective method for modeling receptor–peptide structures should explore a vast conformational binding space, including the large ensemble of conformations explored by the flexible peptide but also the diverse receptor conformational changes triggered by peptide binding. We also hypothesized that maintaining a high level of conformational flexibility or dynamism at the binding interface may be critical for evolving complexes that optimize both peptide recognition and long-range allosteric response, necessitating interactions between multiple functional sites. Hence, to test this hypothesis, we sought to carry out and compare design calculations that either stabilize specific receptor–ligand bound conformations through conformational selection (Fig. 1c, d) or maintain high levels of conformational entropy by enabling the binding of a wide range of peptide conformations (Fig. 1c, e).

Our computational strategy was developed with these ideas in mind and proceeds in the following main steps (Methods, Fig. 1f). Steps (i) to (v) refer to the protein modeling stage while steps (vi) to (xi) refer to the protein design stage.

(i) building hybrid transmembrane (TM) scaffolds in active signaling conformation by homology to diverse chemotactic receptors. Specifically, we construct scaffolds from fragments of receptor structures that have the highest sequence and structure homology to the target receptor using a modified version of the hybridization technique of the software Rosetta[18]. The goal is to generate diverse starting biosensor scaffolds in the active state for subsequent peptide docking. (ii) peptide docking onto the active-state receptor scaffold binding sites generated in step i. The peptide sequence is threaded onto the homologous CX3CL1 peptide structure and the resulting structure is docked through rigid body movements and internal motions around backbone and side-chain dihedrals using the FlexPepDock software[17,19]. The receptor structure is also allowed to move during docking, with side-chains being fully flexible at the receptor–peptide interface. In this step, we aim to identify possible interacting conformations from the large pool of unbound peptide structures. (iii) filter and diversify peptide-bound positions. We developed and applied a diversification method to generate a peptide-bound receptor ensemble representative of the vast conformational binding space and of diverse networks of receptor–peptide contacts. This approach clusters bound peptide conformations, bins the corresponding space and selects representative members of the populated bins that occupy distinct positions in the peptide-binding pocket. (iv) de novo loop rebuilding of the biosensor scaffold. Each peptide-bound receptor scaffold selected in step (iii) is subjected to remodeling of the loops that are in proximity of the bound peptide to best accommodate the peptide conformations. (v) relax the resulting receptor–peptide complex structure. The structures generated in step (iv) are fully relaxed through side-chain repacking and minimization over all conformational degrees of freedom to mimic mutual induced fit effects and identify the most optimal binding conformations. Relaxed structures are clustered and the center of the most populated clusters are selected for the computational design stage.

(vi) computational stabilization of receptor–peptide interfaces through conformational selection (Fig. 1d). Here we apply a classical

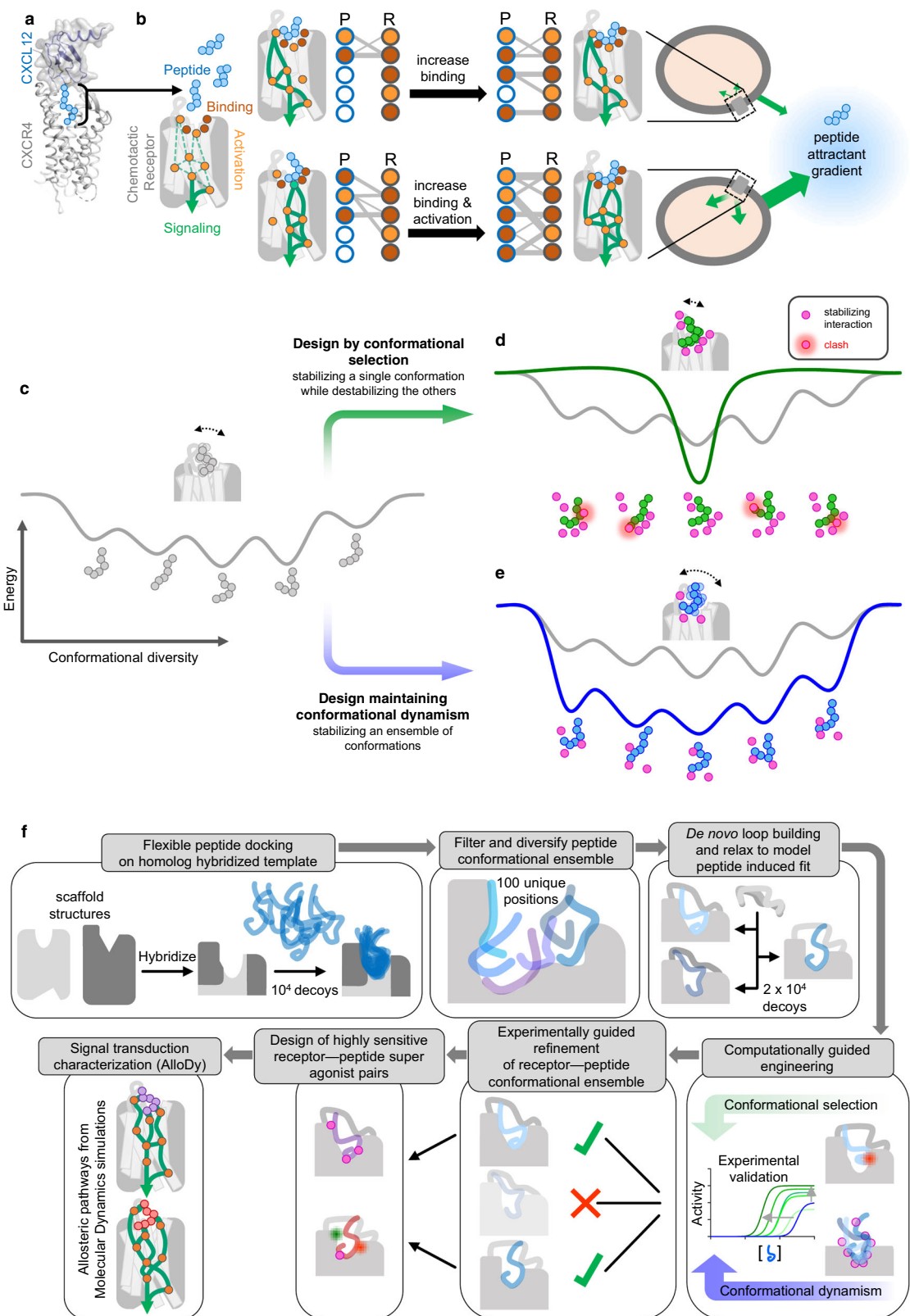

protein binding interface design approach using the software RosettaMembrane to each CXCR4-CXCL12 model selected in step (v). Receptor and peptide residues that are in contact at the binding interface are subjected to substitution to all possible amino acids. A Monte Carlo algorithm randomly selects mutations at multiple sites, predicts, scores the structure of the mutated complex and selects the combination of amino-acid substitutions leading to the largest

decrease in binding energy. (vii) computational design of binding interactions maintaining conformational flexibility (Fig. 1e). Here, designed binding and allosteric contact networks are selected that enhance receptor–peptide interactions in several CXCR4–CXCL12 models, thereby favoring multiple conformations of the complex and maintaining high levels of conformational entropy. This is achieved by designing a library of point mutations at receptor positions in contact

**Fig. 1 | Modeling and design strategy of a Conformationally Adaptive Peptide BioSensor (CAPSen). a** Cartoon representation of the CXCR4–CXCL12 complex (schematic model of receptor and chemokine structures aligned to CXCR2:CXCL8 complex structure. PDB IDs: 4RWS, 4UAI, 6LFO) targeted for the design of chemotactic receptors with enhanced binding and signaling responses towards peptide attractants. The CXCL12 chemokine ligand consists of a folded domain and a flexible 8 residue-long N-terminal tail (sequence: KPVSLSYR represented with light blue spheres) inserted into the receptor binding pocket. **b** The peptide ligand can adopt distinct bound conformations through specific contacts with receptor pocket residues that are classified as drivers of binding (red) or activation (orange). Dotted green lines correspond to putative allosteric signal transduction pathways running through the receptor. Plain green lines schematically represent the specific pathways engaged by each peptide conformation bound to the receptor. Through design, receptor–peptide connectivity (represented as an interaction graph between peptide (P) and receptor (R) residues) can be rewired to promote binding (top), activation, or both (bottom) to ultimately reprogram the cell migratory response. **c**–**e** General overview of the protein-peptide binding design strategies employed in the study. **c** Schematic view of a conformational energy landscape describing the binding of a flexible peptide to a receptor. The peptide is represented with 6 light gray spheres and adopts distinct conformations in each local energy minimum. **d** Design by conformational selection stabilizes one favored receptor–peptide conformation, while destabilizing others. Destabilizing interactions are represented as steric clashes. **e** Design to preserve dynamism selects amino-acid substitutions stabilizing multiple receptor–peptide conformations, hence maintaining conformational entropy at the binding interface. **f** Pipeline of the modeling & design strategy involving receptor–peptide modeling, rational design, experimental validation, refinement of receptor-peptide models, design of peptide super-agonists and analysis of allosteric signal transduction properties.

with the peptide that are compatible with multiple conformations of the complex. (viii) experimental validation of selected designs. The intracellular signaling activities of the engineered biosensors are characterized using cell-based functional assays that report on the activation of Gi which is the main downstream G-protein effector of the CXCR4 receptor; (ix) experimentally guided refinement of receptor–peptide conformational ensemble. We used the experimental validation in step (viii), in particular the peptide potencies that mostly report on binding affinities, to improve modeling accuracy. Specifically, we selected the models with predicted binding energy shifts that best matched the measured shifts in potency. (x) design of highly sensitive receptor–peptide super-agonist pairs. We carried out additional design calculations to search for amino-acid substitutions on the peptide further enhancing binding interactions. (xi) characterization and validation of the allosteric signal transduction properties of the designed complexes. To validate the design calculations and provide mechanistic insights into the signaling properties of the engineered receptor–peptide pairs, we carried out molecular dynamics simulations using our method AlloDy. We analyzed peptide conformational diversity and calculated allosteric signaling transmission properties for each complex.

## Design of hyper-sensitive CAPSens for the native CXCL12 chemokine

As a proof of concept, we modeled and designed peptide ligand agonists starting from the N-terminal partially unstructured agonist region of the chemokine CXCL12 (Fig. 1a), which promotes strong activation of the CXCR4 receptor[20–23]. Specifically, we considered the first 8 N-terminal peptide residues that are the most buried in the CXCR4 binding pocket (Fig. 1a). CXCL12 binding to CXCR4 triggers activation of the receptor, which directly couples to the heterotrimeric G-protein Gi, inhibits the intracellular signaling pathway leading to cAMP production (through the G-protein alpha subunit) and activates that leading to $Ca^{2+}$ release (through the G-protein beta and gamma subunits).

To build and evolve CAPSen scaffolds sensing CXCL12-derived peptides, we selected structural parts from the chemokine receptor family. In absence of a CXCR4 structure in the signaling active form, biosensor templates were assembled from local structures of CXCR4 in the inactive form and the structure of the homologous viral chemokine receptor US28 bound to CX3CL1 (PDB ID: 4XT1)[24], the only active-state chemokine receptor structure available at the time of modeling. Our modeling stage (steps i–v) yielded 9 highly populated clusters of peptide-bound biosensor models. The centers of each cluster was selected as starting templates for the first round of computational design (Methods, Fig. 1c). In the following, we name designs by the approach (Csel for design through conformational selection, Cdyn for design maintaining conformational dynamism, Csedy for combined Csel and Cdyn solutions) and design generation (1 and 2) that refers to a specific round of iterative computational design. Since the first two

N-terminal positions of CXCL12 are critical for activation and even conservative mutations can lead to drastic signaling defects[25–27], we focused our initial computational design on improving the binding of the sensor to positions 3 through 8 of the CXCL12-derived peptide (i.e., P3 through P8), up to the CXC motif.

Our design strategy by conformational selection focuses on the first shell of residues in contact with the peptide ligand (i.e., residues with at least one heteroatom within 4 Å of the peptide), carries out independent combinatorial design calculations on single receptor–peptide conformations and identifies the few complexes with the strongest engineered binding interactions. Complexes are designed using Rosetta's Metropolis Monte Carlo Simulated Annealing protocol[18], scored using the RosettaMembrane energy function[28,29] and selected if they led to decreased interface energies between the receptor and peptide. Using this strategy, we expect to improve binding by selecting a subset of conformations that drive the most potent recognition, thereby decreasing the conformational entropy of the complex (Fig. 1b). The first round of calculations yielded a designed binding hotspot motif with improved interfacial contact density between the TM1/7 interface and P3 of the peptide (Fig. 2a, b). We validated the peptide binding and signaling properties of the Csel1 receptor in HEK cells using cell-based assays reporting G-protein $G\alpha_i$ activation and $Ca^{2+}$ mobilization that are triggered by native chemokine receptors and known to be crucial for chemotactic responses[30,31]. Consistent with the prediction, the designed receptor displayed enhanced sensitivity to CXCL12 (Fig. 2c, d). We built upon the initial success of the Csel1 design by further optimizing the binding interface upstream of P3. A second binding hotspot motif was selected between P7 of CXCL12 and 3 receptor positions lining the β-hairpin of the second extracellular loop (ECL2) (Fig. 2a, b). Combining the 2 designed motifs into the Csel2 receptor led to substantially enhanced potency in calcium mobilization (3.1-fold over WT) and $G\alpha_i$-coupling (3.2-fold over WT) (Fig. 2e, f). Overall, we tested a total of 19 designs from this conformational selection approach and reached a success rate of 37 %.

We then sought to design potent but dynamic receptor–peptide complexes by identifying binding and activating motifs compatible with a wide range of bound conformations, hence maintaining high conformational entropy at the binding interface. We rationally created and screened a computationally guided library of variants built from our initial ensemble of 9 receptor–peptide models. Each variant was designed by mutating a single predicted peptide binding and/or allosteric residue from the first binding shell in the conformational ensemble. Mutations in the library were selected if they did not display significant steric clashes with the peptide in more than 5 of the conformations in the ensemble. A total of 206 point mutant variants were selected in the library and assayed for calcium mobilization (Supplementary Fig. 1) and $G\alpha_i$ coupling (Fig. 2g). Activating point mutations were identified at sites on TM1, TM3, and ECL2 and assembled into the Cdyn receptor variant (Fig. 2a, b; Supplementary Fig. 1). The Cdyn design was considerably more sensitive than the starting CXCR4 WT

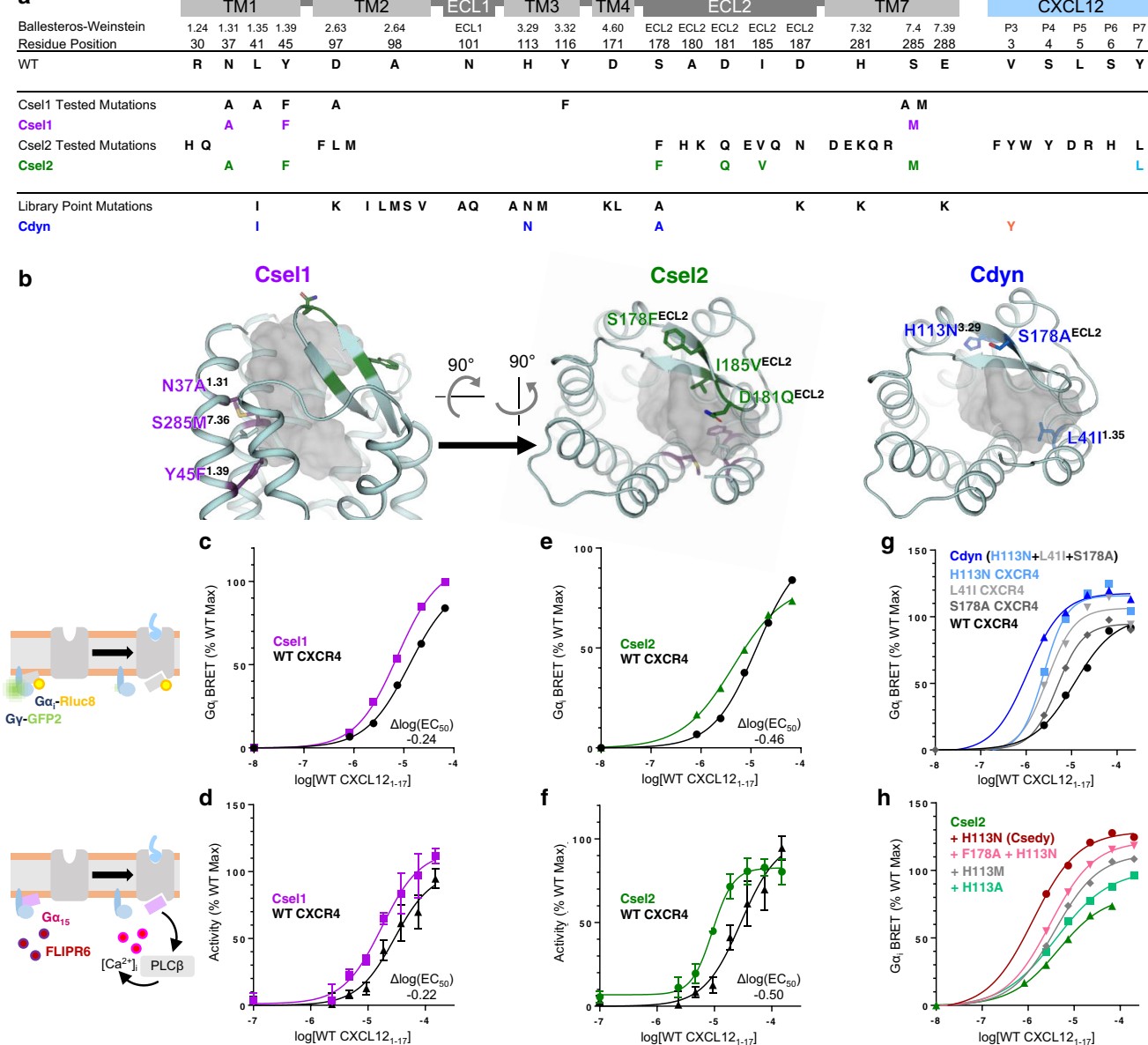

**Fig. 2 | Design of receptor–peptide binding sites for enhanced sensing. a** Table describing the mapping on the receptor topology and numbering of receptor and peptide residues targeted for design. ECL and TM refer to extracellular loop and transmembrane helix, respectively. BW refers to Ballesteros Weinstein notation. Csel1, Csel2, Cdyn represent the sequences of the selected designs. **b** Location of the designed residues (shown in sticks) mapped onto the backbone structure of receptor peptide binding site (shown in cartoon). The WT CXCL12 peptide is represented as a gray-colored surface. **c–h** WT peptide-induced cell signaling responses of designed receptors measured through $G\alpha_i$ activation and calcium release: $G\alpha_i$ BRET of Csel1 design (mean, $n = 2$ technical replicates) (**c**), Csel2 design (mean, $n = 2$ technical replicates) (**e**), and library-screened mutations (mean, $n = 2$ technical replicates) (**g**). Calcium mobilization of Csel1 design (mean ± s.e.m., $n = 3$ technical replicates) (**d**) and Csel2 design (mean ± s.e.m., $n = 3$ technical replicates) (**f**). $G\alpha_i$ BRET of single-point library mutations in Csel2 design background (mean, $n = 2$ technical replicates) (**h**).

scaffold with close to 11-fold enhanced $G\alpha_i$ potency and a 20% increase in efficacy (Fig. 2g). Overall, we fully tested a total of 15 designs from this conformational dynamism approach and reached a success rate of 33%.

We next thought to combine the initially designed binding hot-spot motifs from Csel2 with the binding and activating sites identified in Cdyn. While the positions on TM 1 and 7 strongly overlapped, the substitutions at position 113[3.29] (Ballesteros-Weinstein notation[32]) occupy a region of the binding pocket not exploited by our Csel1 or Csel2 designs. Additive effects were observed when combining the most activating H113N[3.29] mutation with the Csel2 design and led to the Csedy sensor that had the second-most potent and sensitive $G\alpha_i$

response (more than 9-fold increase over WT) against WT CXCL12 in our designs (Fig. 2h).

Overall, these results indicate that our approaches can readily design highly sensitive sensors of the WT CXCL12 chemokine-derived peptide by optimizing both binding and activation determinants. Interestingly, while the Csel designs reached up to 3-fold enhanced potency from WT, Cdyn and Csedy achieved much larger improvements (i.e., up to 11-fold). These differences do not correlate with the number of designed mutations since Csel2, Cdyn and Csedy incorporate 6, 3 and 7 mutations, respectively. Hence, our findings suggest that, compared to the conformational selection approach, the strategy maintaining conformational flexibility has the potential to

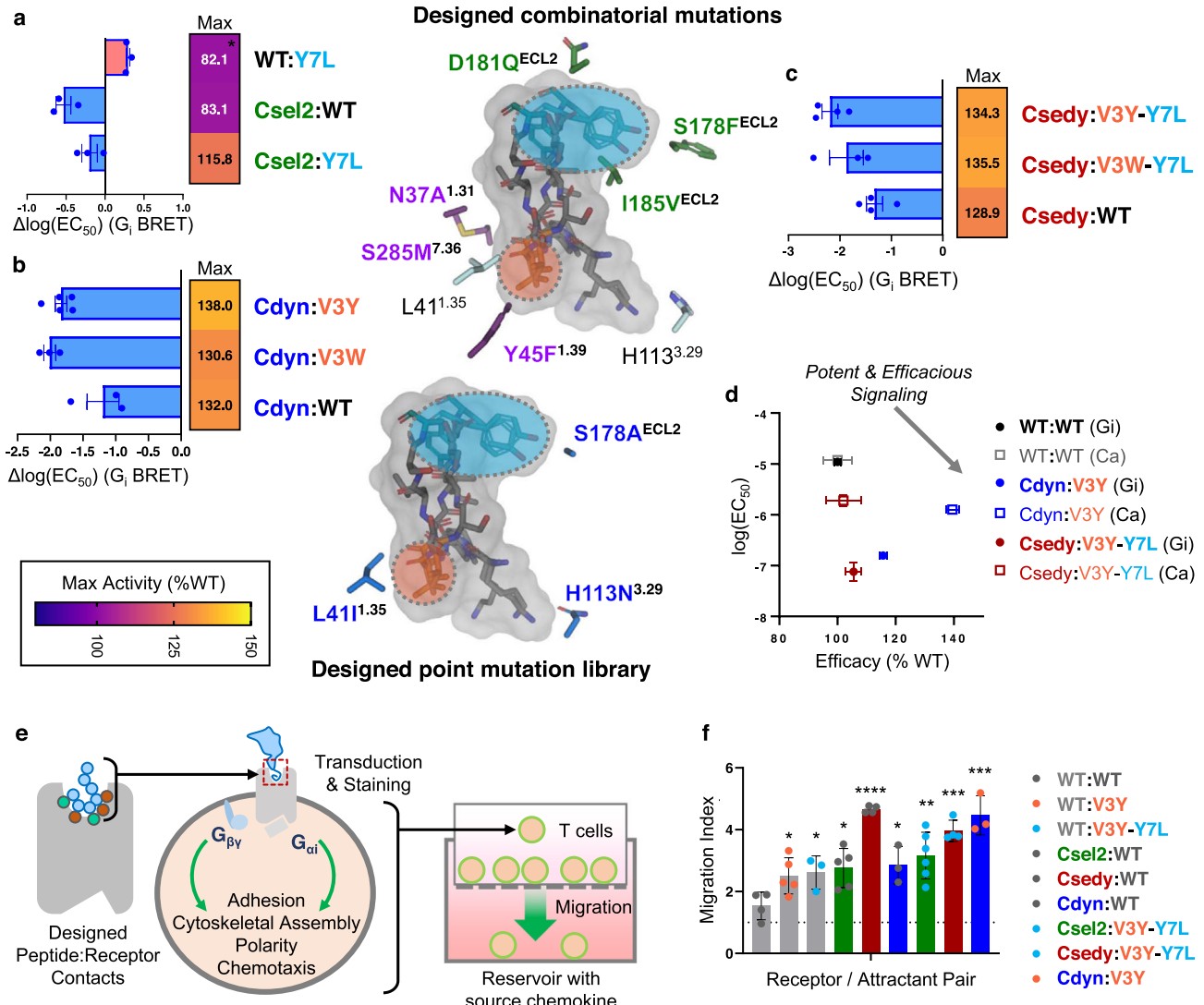

**Fig. 3 | Design of highly sensitive and chemotactic receptor–peptide pairs.**
**a**–**d** Shifts in sensitivity (mean fitted value of dose-response curve fits ± s.e.m., $n = 5$ for Cdyn:V3Y, $n = 4$ for Csedy:V3Y-Y7L and Csedy:WT, $n = 3$ for all other pairs) and maximum activity (fitted value) for various receptor–peptide pairs involving the following designed peptides: **a** CXCL12 Y7L variant, **b** CXCL12 V3 substitutions, **c** CXCL12 V3W-Y7L. **d** Changes in potency and efficacy across three separate experiments (mean ± s.e.m., $n = 3$ independent experiments). **e** Schematic of Boyden chamber migration assay of T cells transduced with engineered receptors and **f** migratory responses of transduced primary human T cells towards full-length chemokine. Bars are colored according to the transduced CXCR4 variant, and individual points are colored according to the CXCL12 variant (mean ± s.d., $n = 3$ for WT:V3Y-Y7L, Cdyn:WT, Cdyn:V3Y; mean ± s.d., $n = 4$ for WT:WT, Csedy:WT, Csedy:V3Y-Y7L; mean ± s.d., $n = 5$ for WT:V3Y, Csel2:WT; mean ± s.d., $n = 6$ for Csel2:V3Y-Y7L). Significance shown with two-sided unpaired t-test $p$ values to WT:WT migration. *$p \le 0.05$, **$p \le 0.01$, ***$p \le 0.001$, ****$p \le 0.0001$ ($p = 0.0297$ for WT:V3Y, $p = 0.0334$ for WT:V3Y-Y7L, $p = 0.0141$ for Csel2:WT, $p < 0.0001$ for Csedy:WT, $p = 0.0205$ for Cdyn:WT, $p = 0.005$ Csel2:V3Y-Y7L, $p = 0.0001$ for Csedy:V3Y-Y7L, $p = 0.0008$ for Cdyn:V3Y).

identify more effective binding interactions that trigger receptor activation.

## Design of CAPSen chemotactic peptide super-agonist pairs

We next sought to create selective receptor–peptide pairs by designing peptide super-agonists. Such synthetic sensor-response systems would provide orthogonal solutions for modulating signal transductions while bypassing the high level of binding promiscuity inherent to native receptors. From our computational models, we identified 2 sites P3 and P7 on the peptide scaffolds where mutations were predicted to further increase binding interactions with the Csel1 and Csel2 receptor designs. A designed Leu at P7 further optimized packing complementarity with the binding hotspot motif of the Csel2 design. This mutation enhanced peptide binding selectivity for Csel2 as Gα$_i$ efficacy of the designed sensor was increased by 30% while the overall response of the WT receptor was decreased by 18% (Fig. 3a). At position P3, our calculations

identified bulky aromatic residues predicted to complement I41$^{1.35}$ and W94$^{2.60}$ at the TM 1/2 pocket interface of the designed sensors, leading to powerful activating effects. Specifically, the Cdyn:V3Y peptide pair displayed more than 80-fold enhanced potency and a 25% increase in efficacy compared to the WT receptor–peptide pair (Fig. 3b, d). The largest signaling increases were obtained when combining peptide mutations at P3 and P7. The Csedy:V3Y-Y7L peptide pair boosted both potency and efficacy by more than 100-fold and 34%, respectively (Fig. 3c, d).

Overall, we observed that large additional enhancements in functional binding properties could be achieved by also evolving the peptide sequence and structures. These results demonstrate the power of our computational approach for engineering synthetic receptor–peptide pairs with highly sensitive binding properties and potent downstream signaling. They also suggest that the binding interface between the native peptide and receptor is far from optimal for binding and signaling potency.

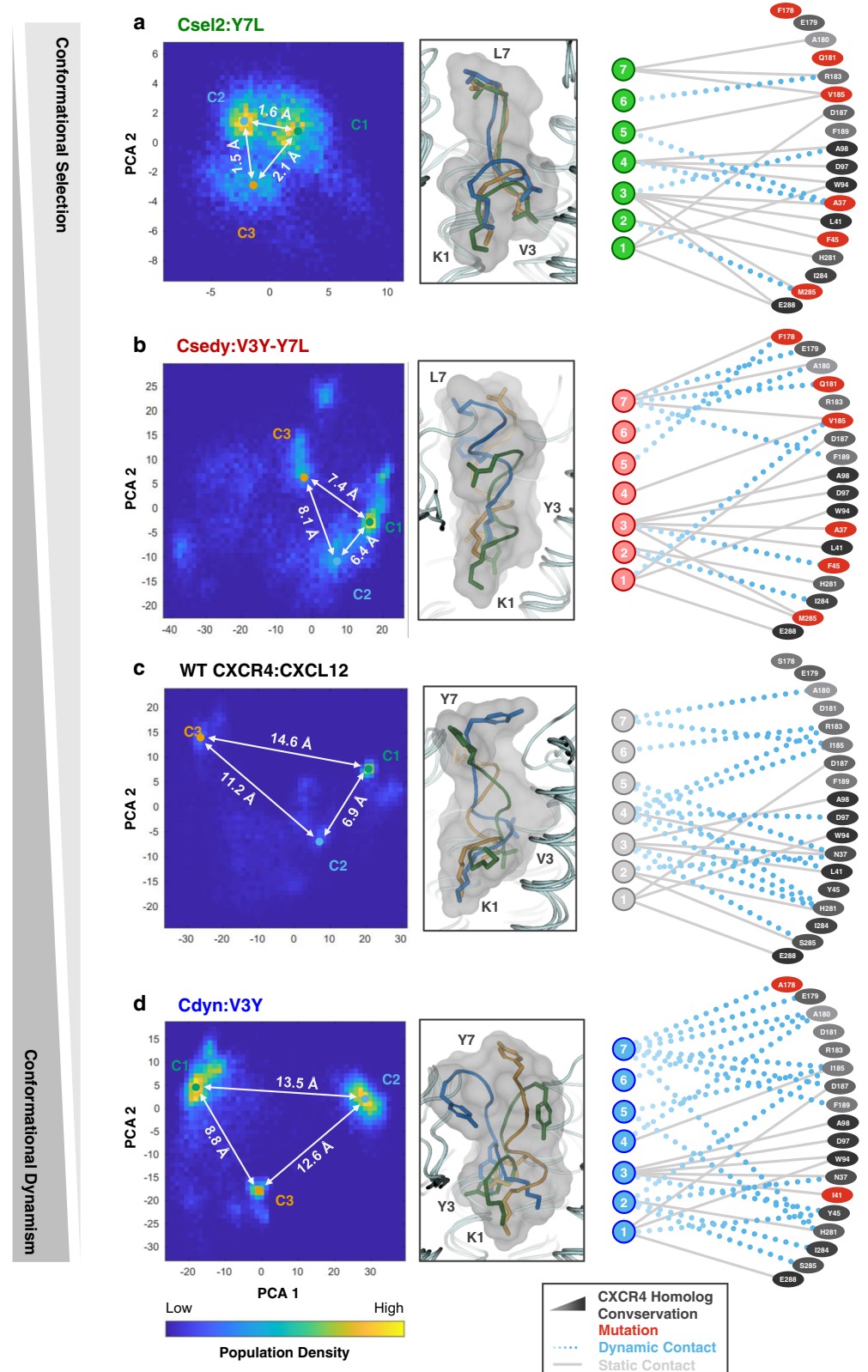

**Fig. 4 | Conformational diversity of WT and CAPSen receptor–peptide substates.** Population density of ligand poses sampled in all-atom MD simulations are plotted in PC space (left panels). Inter-cluster RMSD of the most populated ligand conformations shown. Representative peptide poses are shown for the three most populated conformational substates of each variant (middle panels). Only side-chains for positions 1, 3, and 7 of the peptide are shown for clarity. The top 15 most frequent contacts from each substate are depicted schematically (right panels). Strong static contacts (solid gray lines) are prevalent in two or more substates, while weaker dynamic contacts (dotted blue lines) are unique to a single substate. Receptor–peptide variants are shown in increasing order of conformational dynamism of the complexes: **a** the Csel2:Y7L complex, **b** the Csedy:V3Y-Y7L complex, **c** the WT CXCR4−CXCL12 complex, **d** and the Cdyn:V3Y complex.

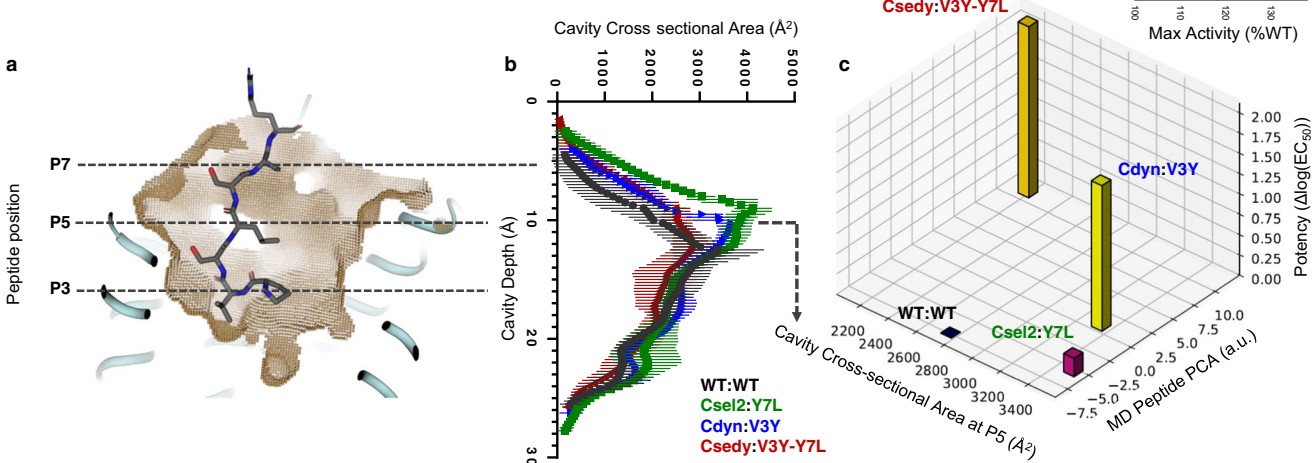

**Fig. 5 | High level of structural adaptation at the designed receptor–peptide binding interface. a** WT CXCR4 ligand-binding cavity with depths marked for N-terminal residues P3, P5, and P7. **b** Cross-sectional area across cluster centers from molecular dynamics simulations. (mean ± s.d.) **c** 3D map of structure-function relationship. Activity shifts from WT of individual receptor–peptide pairs (z-axis for potency and bars colored according to maximal activity) are plotted as a function of conformational shifts of the peptide (y-axis: calculated by Principal Component Analysis on bound peptide ensembles (see Methods)) and conformational shifts of the receptor binding pocket (x-axis: calculated by cross-sectional area at the P5 depth, 10.25 Å (see Methods)) for the center of the largest cluster of conformations.

## Designed receptor–peptide pairs enhanced human T cell chemotaxis

We next assessed whether our ultra-sensitive CAPSens also elicited a migratory phenotype in cells with concomitant sensitivity upon detection of chemokines. Chemotaxis results from the complex orchestration of multiple intracellular pathways that control receptor oligomerization, cell motility, polarity, and adhesion following receptor-mediated G-protein activation triggered by the sensing of chemokine proteins[30,33–35] (Fig. 3e). Such validation represents a stringent test of our ability to leverage molecular design for cell engineering and reprogram complex cellular behaviors in response to environmental cues. We transduced primary human T cells with selected designed sensors (Supplementary Fig. 2) and measured their migration against gradients of full-length WT or engineered chemokines incorporating the designed N-terminal peptide tail. Chemotaxis was measured using Boyden chambers in which cells can migrate across a porous membrane towards a reservoir containing chemoattractant. Migratory indices were measured as the fold-migration over a no-chemoattractant control for each transduced donor. T cells transduced with the Csel2, Cdyn, and Csedy designs displayed up to almost 4.7-fold increased migration towards 100 nM WT CXCL12. At this level of chemoattractant, WT CXCR4 promoted around 1.5-fold enhanced migration when compared to the no-attractant controls (Fig. 3f). Our engineered CAPSens also boosted T cell migration by up to 4.5-fold when exposed to designed chemokines (Fig. 3f). The enhanced cellular migration demonstrates that the designed molecular signaling properties leading to ultra-high sensitivity of our sensors translate into the corresponding modulation of cell functions and phenotypes. It also indicates that our strategy focusing on the flexible peptide region of the chemokine is generalizable to the design of biosensors responding to full-length chemoattractants.

Overall, the in vitro characterization of our designs indicate that the computational techniques have the potential to enrich receptor–peptide binding interfaces for functional interactions that can translate into enhanced receptor signaling sensitivity and potency.

## Highly conformationally adaptive designed receptor–peptide binding interfaces through mutual induced fit

Our designed receptor–peptide agonist pairs offer a unique opportunity to uncover the structural and dynamics underpinnings of receptor–peptide binding and agonism. Experimental structures by X-ray crystallography or cryo-electron microscopy of our designs would only provide snapshots of the conformational ensemble and not reveal whether our designs achieved their functions through the intended modulation of the binding dynamics. Therefore, we decided to instead carry out molecular dynamics (MD) simulations of the designs to investigate the sequence-structure-dynamics relationships underlying their functions. Since the computational design was performed using knowledge-based potentials of the Rosetta software, MD simulations using molecular mechanics force fields provide an orthogonal validation of the design calculations.

Starting from the refined design models that best agreed with the experimental data, we run up to 1.9 microsecond long equilibrium MD simulations in explicit lipids. Within this timescale, the peptide-bound receptor complex remains in the active state as assessed by the local conformation of consensus class A GPCR activation features such as interhelical (i.e., TM3-TM6 and TM3-TM7) distances on the intracellular side of the receptor and the RMSD for the NPxxY motif (Supplementary Fig. 3). However, the simulations are long enough for the peptide and receptor to explore distinct bound conformations and enable a qualitative comparison of dynamic ensembles between variants (Fig. 4, Supplementary Fig. 4).

We analyzed the conformational space of the complexes using principle component analysis (PCA) of the MD trajectories. The analysis revealed the existence of distinct families (i.e., clusters) of conformations for the WT and all engineered receptor–peptide pairs (Fig. 4). The WT and Cdyn binding interfaces were characterized by high peptide ligand RMSD (up to 14.6 Å distance between clusters of peptide conformations, Fig. 4a, b) (Supplementary Table 1) and suggest that the design approach was able to maintain high levels of peptide conformational diversity at the binding interface. On the other hand, the Csel2 design displayed substantially lower conformational diversity (only up to 2.1 Å inter-cluster RMSD, Fig. 4c), consistent with that design strategy stabilizing a subset of the receptor–peptide structures through conformational selection. As expected for the hybrid design strategy, the V3Y-Y7L peptide explored an intermediate conformational space in the binding pocket of Csedy (up to 8.1 Å, Fig. 4d). Overall, the different levels of peptide structural heterogeneity identified by the MD simulations are consistent with the intended modulation of the conformational space by our 2 design strategies. To rule out that the observed conformational heterogeneity solely results from potential inaccuracies in our predicted models, we carried out the same MD analysis starting from the experimental

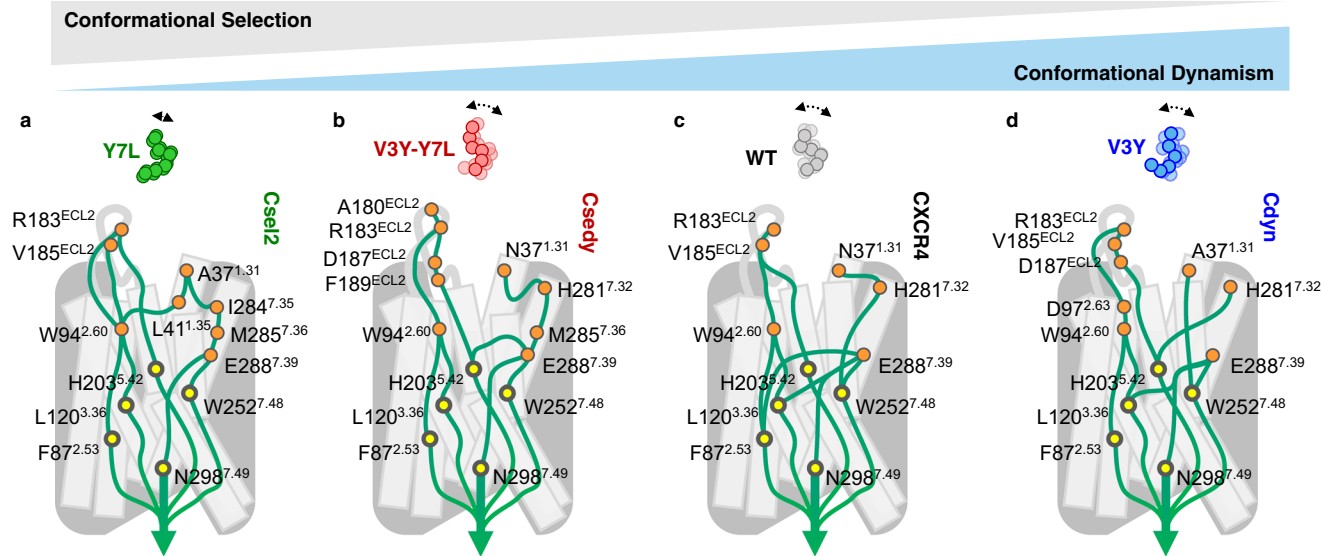

**Fig. 6 | Overview schematic of the allosteric signal transduction in CAPsen designs.** Signal transduction pathways calculated by AlloDy from peptide-contacting residues (orange circles). The allosteric pathways utilize a diversity of receptor pocket residues to communicate information, but ultimately propagate the activation signals through a common set of allosteric determinants in the core of the receptor (yellow circles: F87$^{2.53}$, L120$^{3.36}$, H203$^{5.42}$, W252$^{6.48}$, N298$^{7.49}$). (See Figs. S7–S10 for a complete mapping of the allosteric couplings.) Receptor–peptide variants are shown in increasing order of conformational dynamism of the complexes: **a** the Csel2:Y7L complex, **b** the Csedy:V3Y-Y7L complex, **c** the WT CXCR4–CXCL12 complex, and **d** the Cdyn:V3Y complex.

structure of the related complex between the N-terminal peptide of RANTES[36] and the CCR5 chemokine receptor. We observed a similar diversity in the conformations of the bound chemokine peptide (Supplementary Fig. 5), suggesting that native chemokine receptors may actually bind agonist peptides with a significant degree of conformational dynamism.

We then analyzed in detail the network of binding contacts engaged by the distinct families of peptide conformations. Contacts were defined as dynamic if they were unique to one cluster or static if they were observed for at least two peptide binding modes. Throughout the MD trajectories, the peptide engaged with Csel2 through 16 strong static versus 5 weaker dynamic binding contacts (Fig. 4). The number of static contacts dropped to 12 and 11 while the dynamic ones raised to 11 and 13 in the Csedy and Cdyn complexes, respectively (Fig. 4). These observations further confirm that Cdyn and Csedy complexes involve a more dynamic binding interface than Csel2.

Conformational diversity was also noticeable on the receptor side and best quantified using a volumetric analysis of the peptide binding pocket. To simplify the analysis, representative members of each cluster were selected and cross-sectional areas were calculated at different depths of the binding site. This analysis highlighted significant conformational adaptation of the binding surface (e.g., by up to 52% at a cavity depth of 10.25 Å) in response to the different peptide conformations and sequences (Fig. 5a, b). When we mapped the distribution of the largest cluster of conformations (i.e., cluster C1) onto a 3D map of the structure-function relationship (Fig. 5c), we observed that the designed pairs occupy subspaces that are far apart in both receptor binding pocket and peptide conformation dimensions.

Overall, these findings suggest that the high conformational plasticity of the CXCR4–CXCL12 binding interface may facilitate the adaptation of contact networks in response to even limited changes in receptor and peptide sequence space through mutual induced fit. Although this analysis implies that higher conformational flexibility at the binding interface correlates with stronger signaling efficacy, it does not provide mechanistic insights into how such structurally distinct binding complexes could trigger potent signaling responses.

## Potent signaling achieved through substantially rewired but robust allosteric pathways

To address that question, we sought to investigate how peptide binding initiates signal transductions across the receptor. Since the inference of allosteric pathways using experimental approaches remains very challenging and would require extensive measurements by NMR spectroscopy, we relied on predictions from MD simulations. Allosteric signal transductions are thought to be primarily mediated by networks of dynamically coupled residues[37–40]. Hence, they should be identifiable through analysis of coupled motions extracted from MD simulations of peptide-bound receptor complexes in the active state. We developed the method AlloDy to carry out this analysis (Methods, Supplementary Fig. 6). First, AlloDy calculates the amount of information (i.e., Mutual Information (MI)) exchanged between residues from their levels of coupled motions, a metric adopted by other approaches as well[38]. Then the method constructs all possible allosteric networks connecting distant sites in the receptor and running through residues exchanging high MI. These individual paths are then clustered into allosteric pipelines that describe how allosteric signals are propagated from the extracellular peptide-binding pocket to the intracellular G-protein coupling site (Methods, Fig. 6). Importantly, since our MD simulations are performed on peptide-bound receptor complexes in the active state, they do not carry out information on the transition of the receptor from inactive to active states, but inform on how the extracellular and intracellular sites communicate when the peptide is bound and the receptor occupies the active state. Within that framework, effective signal transductions should translate into strong allosteric pipelines running through the receptor structure and connecting the intra- and extracellular receptor sides.

As shown in Fig. 6, AlloDy identified several allosteric pipelines for the WT CXCR4 and designed CAPSens that flowed from the agonist peptide through a layer of receptor residues at the pocket interface, termed 'allosteric triggers', down to a conserved set of 'allosteric transmission hubs' located in the TM region away from the shell of ligand-binding residues. These transmission hubs include highly conserved class A GPCR motif residues W252$^{6.48}$ (W toggle) and N298$^{7.49}$ (NPxxY), as well as key highly conserved activation residues in CXCR4: F87$^{2.53}$, L120$^{3.36}$, H203$^{5.42}$, mutation of which has been shown to impair

calcium mobilization activity.[41–43] Though the relative amounts of information passing through this conserved set of transmission hubs varies between the designs (Supplementary Table 2), the topology of the allosteric pipelines leading to G-protein signaling remain very similar between variants (Supplementary Figs. 7–10).

By contrast, how allosteric signals initiated by the peptide reach the common set of transmission hubs is highly dependent on the designed receptor–peptide interface. In fact, each variant utilizes a distinct set of allosteric triggers to connect to the activating allosteric pipelines (**WT**: R183$^{ECL2}$, I185$^{ECL2}$, W94$^{2.60}$, N37$^{1.31}$, H281$^{7.32}$, E288$^{7.39}$, see Supplementary Fig. 7; **Csel2**: R183$^{ECL2}$, I185$^{ECL2}$, W94$^{2.60}$, A37$^{1.31}$, L41$^{1.35}$, I284$^{7.35}$, M285$^{7.36}$, E288$^{7.39}$, see Supplementary Fig. 8; **Cdyn**: R183$^{ECL2}$, I185$^{ECL2}$, D187$^{ECL2}$, D97$^{2.63}$, W94$^{2.60}$, N37$^{1.31}$, H281$^{7.32}$, see Supplementary Fig. 9; **Csedy**: A180$^{ECL2}$, R183$^{ECL2}$, V185$^{ECL2}$, D187$^{ECL2}$, F189$^{ECL2}$, W94$^{2.60}$, A37$^{1.31}$, H281$^{7.32}$, I284$^{7.35}$, M285$^{7.36}$, E288$^{7.39}$, see Supplementary Fig. 10). Though there is some overlap in the allosteric triggers, the specific interactions with the peptide agonist that confer activation are unique across WT and the CAPSens and reflects the high diversity of binding contacts engineered at the binding interface.

Overall, while there is substantial dynamic rearrangement of activating contacts between variants (Fig. 6), they propagate the activation signals through a common set of allosteric determinants in the core of the receptor. These remarkable findings suggest that high conformational adaptability of the binding site together with robustness of the allosteric transmission layer are critical features for the evolution of signaling receptor–peptide pairs.

## Discussion

Current approaches for the design of protein-protein complexes mostly follow the classical lock and key paradigm and optimize interactions between static binding surfaces. The lack of dynamic treatment of protein interactions hampers the effective design of complexes involving flexible proteins and ligands, which represent a large fraction of the molecules regulating cellular functions. In particular, peptide-binding receptors constitute one of the most abundant signaling systems in humans. However, these complexes have been particularly challenging to study and engineer partly owing to their high conformational flexibility. Here, we developed a computational framework for the design of dynamic protein complexes involving adaptable conformational ensembles of the molecules. We demonstrate that this approach is critical to achieve optimal sensing of flexible ligands and strong allosteric signaling responses that necessitate the interactions with multiple functional sites on the receptor surface.

Previous efforts to engineer high-affinity peptide ligands to the native CXCR4 receptor have mostly generated antagonists. Library-selected CXCL12-based antagonists[44] identified single N-terminal amino acid additions of Leu and Met that may antagonize CXCR4 in a similar mode as vMIP-II[45], IT1t[46] or Met-CXCL12[47], that intercalate between W94$^{2.60}$ and H113$^{3.29}$, a local hydrophobic inter-TM groove not occupied by our agonist peptide ensemble. These antagonists may lock W94$^{2.60}$ in an inactive conformation, preventing allosteric signal propagation through key downstream transmission hubs, suggesting that selecting solely for high-affinity receptor–peptide interactions is susceptible to conformational selection of inactive receptor states.

Our designs were able to explore regions of sequence space not enriched by such binding-selective approaches, suggesting that the computational method can explore and engineer alternative active states not commonly accessed by the WT receptor. Unlike most binding interfaces between globular proteins, our designs displayed considerable structural adaptation to sequence changes. Remarkably, the diversity of designed allosterically coupled residue networks at the ligand-binding pockets is large among variants, and even between conformational substates of the same variant. While the designed allosteric triggers still operate and funnel signals through the same set of conserved transmission hubs as WT, they considerably enhance

signal transduction through optimally rewired dynamic couplings. Our CAPSens are capable of ultrasensitive responses, not by enriching a high density of strong contacts around a particular active conformational substate, but by preserving conformational dynamism, as observed in our MD simulations. Our findings support a receptor–peptide recognition model where conformational flexibility is essential for the bound molecules to engage a multitude of functional interactions triggering effective allosteric responses. In this model, high levels of conformational entropy enable the shifting of active state ensembles and the rewiring of allosteric coupling via contacts not commonly accessed by the WT complex (Fig. 6, Supplementary Figs. 7–10). As such, the high conformational adaptability of the native CXCR4–CXCL12 binding interface is critical in accommodating and rewiring allosteric entry points to the transmission layer. Overall, our study suggests that the combination of a flexible sensing layer coupled with a robust signal transmission layer may be a common hallmark of GPCRs, providing potential mechanistic insights into the high evolvability of sensing and signaling properties in this receptor family. While our computational findings are consistent with the experimental results, the MD simulations were performed on design models and not experimental structures. Hence, we cannot rule out that the precise details of the simulated binding complexes may be affected by inaccuracies in the starting structural models.

This work was started before AlphaFold2 was released. A recent study[48] indicates that AlphaFold can predict peptide–protein interactions despite not being trained on this task, suggesting that the main features of peptide binding can be implicitly captured as an extension of folding. Our method is geared towards modeling flexible peptide interactions which do not involve strong patterns of unique static contacts such as those characteristic of folded polypeptide chains. Therefore, our study should provide a complementary and useful approach to neural network based methods trained on protein folded structures.

In the long run, we expect that designed chemotactic signaling systems should prove useful in a wide variety of therapeutic contexts. Chemotactic peptides are attractive targets since directional movement of cells in response to gradients of these molecules (i.e., chemotaxis) is essential throughout biology and control over cell migration represents a key challenge in synthetic cell biology. For example, efficient immune cell homing to and into cancers is one of the main bottlenecks in modern immunotherapy[49–53]. Hence, these therapeutic approaches would benefit from engineered cytotoxic lymphocytes with enhanced chemotaxis toward tumor sites. Overall, our results suggest that engineered receptors could trigger migration towards cancer-prone sites at longer distances with shallower chemokine gradients when compared to native chemotactic systems. Our designed CAPSen:hyper-agonist peptide pairs open the door to bringing cell migration under exogenous and spatiotemporal control, providing a promising synthetic cell biology tool.

Most biosensor design approaches have focused on engineering protein domains for optimal recognition of structurally well-defined molecules. Previous studies have repurposed designer receptors exclusively activated by a designer drug (DREADDs) to elicit chemotaxis towards the small molecule clozapine-N-oxide (CNO)[54], but the direct in vivo application of this approach is limited by the delivery of CNO and the inherent lack of utility as a gradient-generating homing molecule. Our CAPSens exhibit some degree of orthogonality in the Csel2:Y7L pair, and future iterations could be developed as genetically encodable orthogonal receptor–peptide pairs allowing for biological expression of the homing signal by cells that would enable synthetic transmitter-receiver cell systems and precise spatiotemporal control of cell homing. By targeting flexible and structurally uncharacterized peptides, our design platform significantly expands the range of molecules that can be detected by biosensors. Unlike approaches that rely on multi-domain sensor reconstitution upon ligand sensing, our

method optimizes the coupling between molecular recognition and allosteric response in a single protein domain within the restricted design space of the ligand pocket interface and can generate CAPSens with strongly enhanced dynamic and sensitive responses. Carving biosensors into versatile GPCR scaffolds offers key additional advantages. GPCRs can now be engineered to trigger a wide range of intracellular functions through reprogrammed coupling to diverse effectors including G-proteins and arrestins[55,56]. Alternatively, inserting fluorescent protein domains into GPCR scaffolds enables fast and direct optical detection of ligand molecules[57]. As such, our approach lays a foundation for a wide range of synthetic biology, diagnostics, and therapeutic applications that would benefit from sensor systems that trigger complex cellular outputs or enable direct highly sensitive detection of chemical cues.

## Methods

### Constructing initial chemokine biosensor scaffolds

Our initial goal was to build CXCR4-based receptor scaffolds in the active signaling state for engineering precise interactions with peptide agonists that promote strong binding and potent response. In absence of a CXCR4 active state structure, hybrid scaffolds were generated using elements from the inactive-state structure of CXCR4 crystallized with a viral chemokine antagonist (PDB ID: 4RWS)[45], and the active-state structure of the viral chemokine US28 receptor crystallized with CX3CL1 (PDB ID: 4XT1)[24]. The hybridization aimed to incorporate the maximal number of active state structural features from 4XT1 while preventing significant de novo reconstruction of the transmembrane core region due to poor sequence-structure alignment between the viral chemokine template and the CXCR4 sequence. Hybridized scaffolds incorporated structural elements of either 4RWS or 4XT1 around the peptide binding pocket in ECL2 (residues 174$^{4.63}$–192$^{ECL2}$) and the extracellular head of transmembrane helix (TM) 2 (residues 87$^{2.53}$–101$^{ECL1}$), local regions that differ significantly between both templates (Fig. 1f). Combination of these structural elements from both structures generated 4 receptor scaffolds originating from the active-state homologous template of US28 using either, both or neither of the ECL2 or TM2 structural elements from 4RWS. Subsequently, the CXCR4 sequence was threaded onto the hybrid template structures guided by sequence-structural alignments using HHpred (US28 and CXCR4 share 29 % sequence identity overall)[58,59]. Before hybridization, the 4RWS template structure was corrected back to wild-type sequence (C187D$^{ECL2}$ / W125L$^{3.41}$) and missing loops in ICL1 and the truncated ICL3 were repaired by the loop remodeling protocol of Rosetta (cyclic coordinate descent algorithm)[60,61]. 1000 decoys were generated and clustered and the lowest energy cluster was selected for further modeling.

### Peptide docking

The N-terminal tail of CXCL12 in experimental structures of the chemokine is often too disordered or lacks receptor context to truly represent an active-state conformation. Thus, we threaded the sequence of that region onto the active-state structure of CX3CL1 in complex with US28 (4XT1) to generate a starting template for subsequent flexible docking. The N-terminal 11 residues of CXCL12, including the CXC motif, were threaded onto the CX3CL1 peptide structure from the US28:CX3CL1 active-state complex (PDB ID: 4XT1). The N-terminal K1 of CXCL12 was aligned to the H2 position of CX3CL1 to match the partial positive charge of the imidazole ring since the Q1 residue of CX3CL1 is cyclized to form pyroglutamate to produce a neutral N-terminus[24]. (Alternately, K1 of CXCL12 was aligned to H3 of CX3CL1, but docking from this initial position yielded models with weak interface energies and few contacts to key binding residues). That initial peptide position was translated across the receptor pocket in a cubic grid around the aligned position and rotated around the principal axis of the receptor at each lattice point to evenly cover the space of the receptor binding cavity. Input poses are prepacked to generate 116 starting positions for subsequent flexible peptide docking using the Rosetta FlexPepDock protocol[17]. In these simulations, the peptide is docked using a Metropolis Monte Carlo simulated annealing protocol involving iterative cycles of peptide rigid body moves and peptide backbone structure optimizations, followed by side-chain repacking and minimization over all conformational degrees of freedom of the peptide residues and peptide-contacting receptor residues. 10,000 decoys were generated from the 116 unique starting inputs. 8 separate flexible peptide docking simulations (10,000 decoys each) were run from each of the 4 starting scaffolds (320,000 total decoys) using no constraints or one of 7 different sets of constraints were used to enrich the following putative receptor–peptide interactions that represent known critical agonistic contacts (CXCR4 residue to CXCL12 residue)[41,62,63]:

Constraints follow the format:

AtomPair <atom><residue> FLAT_HARMONIC <distance><sd> <tolerance>

The constraint format describes a harmonic distance penalty between two atoms with a zero penalty in the range of <distance – tolerance> to <distance + tolerance>. The penalty is harmonic with width parameter <sd> outside of that range:

$$f(x) = \left( \frac{x - (distance + tolerance)}{sd} \right)^2 \tag{1}$$

One of the following seven constraint sets were applied for separate production runs from each of the four receptor scaffolds, generating 10,000 decoys in each run.

(1) D97 to the ε-amine of K1:
AtomPair CG 71 NZ 296 FLAT_HARMONIC 1.75 0.2 1.75
(2) D97 to S4:
AtomPair CG 71 OG 299 FLAT_HARMONIC 1.75 0.2 1.75
(3) D171 to the ε-amine of K1:
AtomPair CG 145 NZ 296 FLAT_HARMONIC 1.75 0.2 1.75
(4) E288 to the ε-amine of K1:
AtomPair CD 262 NZ 296 FLAT_HARMONIC 1.75 0.2 1.75
(5) E288 to the N-terminal amine:
AtomPair CD 262 N 296 FLAT_HARMONIC 1.75 0.2 1.75
(6) Tripartite constraint set for D97 to S4 + E288 to the ε-amine of K1 + D171 to the N-terminal amine:
AtomPair CG 71 OG 299 FLAT_HARMONIC 1.75 0.2 1.75
AtomPair CD 262 NZ 296 FLAT_HARMONIC 1.75 0.2 1.75
AtomPair CG 145 N 296 FLAT_HARMONIC 1.75 0.2 1.75
(7) Tripartite constraint set for D97 to S4 + D171 to the ε-amine of K1 + E288 to the N-terminal amine:
AtomPair CG 71 OG 299 FLAT_HARMONIC 1.75 0.2 1.75
AtomPair CG 145 NZ 296 FLAT_HARMONIC 1.75 0.2 1.75
AtomPair CD 262 N 296 FLAT_HARMONIC 1.75 0.2 1.75

### Peptide conformation selection through diversification

For each production run of unconstrained or constrained peptide docking on one of the initial receptor scaffolds, the docked peptide decoys were filtered by a combined interface and peptide energy score (Rosetta score terms "I_sc + pep_sc" from the FlexPepDock protocol). The decoys were aligned along the receptor structures and the $C_\alpha$ coordinates ($C_x$, $C_y$, $C_z$) of each of the 20% top-scoring poses were stored in a matrix, whose principal axes were calculated in reference to the common principal axes of the receptor. Each peptide pose was described by three features:

(i) peptide position $P_p$ relative to the receptor by center of mass of the peptide:

$$P_p = \sqrt{C_x^2 + C_y^2 + C_z^2} \tag{2}$$

(ii) peptide orientation $P_o$ by the angles of the peptide's 3 principal axes (1,2,3) in reference to the 3 principal axes of the receptor (a,b,c):

$$P_o = \sqrt{\theta_{1a}^2 + \theta_{2b}^2 + \theta_{3c}^2} \qquad (3)$$

(iii) internal peptide shape $P_s$ by the eigenvalues of the (3 ×11 residue) dimensioned matrix of the $C_\alpha$-coordinates of the peptide

$$P_s = \sqrt{e_1^2 + e_2^2 + e_3^2} \qquad (4)$$

Peptide poses which were more than 3 standard deviations from the average value for any of the 3 variables were removed. The remaining structures were sorted into N bins by $P_p$, then each of those N bins were sorted into N bins by $P_o$, and finally into N bins again by $P_s$, such that ~200 populated bins were identified with unique $P_p$, $P_o$, and $P_s$ properties. For each bin, the decoy with the best combined interface and peptide energy score was selected for loop rebuilding and relaxation.

## Loop rebuilding and complex relaxation to model mutual induced fit effects

Loop rebuilding and complex relaxation were performed on each diversified ensemble of peptide poses from flexible peptide docking to generate ~20,000 decoys per receptor scaffold and constraint combination for a total of 640,000 decoys.

*Loop rebuilding*: Extracellular loops either truncated in the starting scaffold or in proximity to the bound peptide were rebuilt using the loop remodeling protocol of Rosetta. 100 independent trajectories were started from each peptide-bound receptor structure selected by the diversification step.

*Complex relaxation*: To further capture and model peptide induced fit effects on the receptor structure, receptor–peptide complexes with closed loops coming out of the loop rebuilding step were subsequently relaxed across all conformational degrees of freedom. All receptor structures were restrained using distance constraints derived from sequence conservation[56]. Any of the 7 experimentally derived receptor–peptide interface constraint sets used in peptide docking (above) were again applied during complex relaxation. For each combination of initial receptor scaffold and peptide docking constraints (~20,000 decoys each), the 10 % lowest interface energy decoys were clustered by structural similarity of the peptide conformation and key binding residues in the receptor pocket. The most populated clusters (i.e., containing at least 30 members) were filtered by a combined interface and peptide energy score (Rosetta score terms "I_sc + pep_sc" from the FlexPepDock protocol, <−45 REU). Top-scoring clusters of models were also filtered for interfacial constraint satisfaction (<10 REU constraint violation penalty, which ensures constrained atoms were within 4.1 Å). Because the 2 N-terminal residues of CXCL12 have been shown to be essential for activity, clusters that did not display any contacts between K1 or P2 to key receptor residues known to be important for activity were not considered. A total of 9 clusters passed these filtering steps and the representative models from these clusters were selected for the design of receptor–peptide complexes.

## Refinement of the ensemble of CXCR4–CXCL12 active-state models

While the initial set of WT CXCR4–CXCL12 models provided key input scaffold structures for engineering functional receptor–peptide interactions, we did not expect every model to accurately represent the receptor–peptide conformational ensemble. We filtered and refined the initial set of models to find an ensemble of flexible peptide dock positions that best recapitulate the observed mutational effects and increase overall prediction accuracy. Because all mutations were designed at positions that directly contact the peptide, we hypothesized that

changes in ligand $EC_{50}$ should be dominated by changes in ligand binding affinity. Therefore, the change in binding interface energy from WT of each model was compared to the $EC_{50}$ shifts. Since the designed mutations triggered large increases in potency, we selected the models that displayed substantial decrease in binding interface energies (i.e. from −3.9 Rosetta Energy Units for Csel2 to −6 Rosetta Energy Units for Cdyn). The models that best supported of observed changes in $EC_{50}$ for each variant were used as an initial input for further flexible peptide docking refinement to identify optimal conformations for the WT, Csel2, library-selected and Csedy designs. A single constraint was enforced for electrostatic interaction between E288[7.39] of CXCR4 and the ε-amine of K1 on the peptide. Top-scoring cluster members then underwent another round of side-chain repacking and energy minimization without constraints. The interface energies of the resulting models were again validated against observed changes in $EC_{50}$ to identify conformational states representative of an ensemble of peptide positions which largely support the designed effects measured experimentally.

## Computational combinatorial design

Designable sites were identified on both the peptide and receptor sides of the different binding interfaces featured in the initial set of 9 CXCL12-bound receptor WT models. Designed combinations of amino-acids and conformations were searched concurrently for improving receptor–peptide association and signaling response. In silico mutagenesis was performed as previously described[64], allowing all 20 possible residue substitutions at designable sites and selecting top-scoring models for interface energy improvement from WT among 200 independent trajectories, such that scores converged for the top 10% models. All residues with heteroatoms within 5.0 Å of any designable residue were repacked and their backbone and side-chain minimized. Csel2 designs were made on two different clusters of models that showed good agreement with the initial Csel1 design. Designs were computationally validated by peptide docking refinement (10,000 independent trajectories) to identify the optimal docked peptide position at the binding interface of the designed complexes and refine the binding energy predictions. The 10% lowest energy decoys were verified by RMSD to the intended design position, cluster size, and interface energy after repacking.

## Computationally guided point mutant library

A computationally guided library of variants was built from the initial ensemble of receptor–peptide models. Each variant was designed by mutating a single predicted peptide binding and/or allosteric residue. The mutant library consisted of substitutions involving modest changes in side-chain size, and polarity that would largely be compatible with the ensemble of initial receptor–peptide conformations. Specifically, mutations were selected if they did not display significant steric clashes with the peptide (>4 Rosetta energy Units) in more than 5 of the conformations in the ensemble. The following mutations were included in the library: R30A/K/Q/L/I/M[1.24], N33A/Q/V/L/I/M[1.27], A34S/V/L/I/M[1.28], N37A/Q/V/L/I/M[1.31], L41I/V/F/M[1.35], Y45A/F/L/I/M/W[1.39], W94A/Y/F/L/I/M[2.60], D97E/N/V/L/M/K[2.63], A98S/V/L/I/M[2.64], N101A/Q/V/L/I/M/K/R[ECL1], H113A/N/Q/T/V/F/L/I/M/Y[3.29], Y116A/L/I/M/W[3.32], D171A/E/N/V/L/I/M/K[4.60], S178A/T/V/L/I[ECL2], A180S/V/L/I/M[ECL2], D181A/E/N/V/L/I/M[ECL2], D182A/E/N/V/L/I/M[ECL2], R183A/K/Q/L/I/M[ECL2], I185A/L/F/M[ECL2], D187A/E/N/V/L/I/M/K[ECL2], R188A/K/Q/L/I/M[ECL2], F189A/Y/L/I/M/W[ECL2], Y190A/F/L/I/M/W[ECL2], V196A/T/L/I/M[5.35], Q200A/N/V/L/I/M[5.39], H203A/N/Q/T/V/F/L/I[5.42], Y255A/F/L/I/M/W[6.51], I259A/L/V/F/M[6.55], D262A/E/N/V/L/I/M[6.58], H281A/N/Q/T/V/F/L/I/K[7.32], I284A/L/V/F/M[7.35], S285A/T/V/L/I[7.36], E288A/D/N/V/L/I/M/K[7.39], F292A/Y/L/I/M/W[7.43].

## Analysis of the conformational dynamics and allosteric signaling properties of the receptor–peptide complexes

To cross-validate the designed signaling properties of the receptor–peptide complexes, we developed AlloDy, a method

predicting allosteric signal transduction pathways from molecular dynamics simulations. As described in detail below, the approach first runs all-atom simulations of the complexes in explicit solvent, extracts distinct peptide-bound receptor conformations from the simulated ensemble using Principle Component Analysis (PCA), calculates mutual information (MI) from inter-residue correlated motions in each conformation and identifies allosteric pathways that maximize the MI transferred from the ligand to the G-protein binding sites.

## Molecular dynamics (MD) simulations

The final selected models for CXCR4: WT:WT, Cdyn:V3Y, Csel2:Y7L, and Csedy:V3Y-Y7L complexes were used as starting input poses for MD simulations. CCR5-RANTES simulations were started from the 11 N-terminal residues of RANTES bound to the receptor extracted from the active state structure of the complex (PDB: 7F1R). The receptor-ligand complex was inserted into a regular hexagonal POPC lipid bilayer with 90 Å perpendicular distance between any parallel sides and solvated by 22.5 Å layer of water above and below the bilayer with 0.15 M of Na$^+$ and Cl$^-$ ions using CHARMM-GUI bilayer builder[65,66]. Simulations were performed with GROMACS 2020.5[67,68] with CHARMM36 forcefield[69] in an NPT ensemble at 310 K and 1 bar using a Nosé−Hoover thermostat (independently coupled to three groups: protein, membrane, and solvent with a relaxation time of 1 ps for all three) and Parrinello-Rahman barostat (with semi-isotropic coupling at a relaxation time of 5 ps respectively). Equations of motion were integrated with a timestep of 1 fs for the first three steps of equilibration and then 2 fs using the leap-frog algorithm. Each system was energy minimized using the steepest descent algorithm for 5000 steps, and then equilibrated with the atoms of the ligand-receptor complex and lipids restrained using a harmonic restraining force in 6 steps (Supplementary Table 3). After constrained equilibration, 5–7 independent trajectories of 200 or 300 ns (Supplementary Table 4) were run for each system. The first 50 ns of the simulations were discarded as the time needed for the system to equilibrate, as shown by the Cα RMSD of the receptors and the ligands. The total simulated time was defined to ensure convergence of the 1st and 2nd order entropies calculations in the top three PCA clusters in every system (see sections below).

## Principle component analysis (PCA) of bound peptide conformational ensemble

All MD trajectories sampled the receptor active state as assessed by the structural distribution of consensus class A GPCR activation features such as the TM3-TM6 and TM3-TM7 interhelical distances on the intracellular side of the receptor, except for the C2 and C3 substates of the variant Cdyn for those we also observed a minor alternative minimum not representative of a true active state (i.e., distinct from the well containing the experimental active state structures). The frames corresponding to these minor populations were filtered out to ensure that subsequent conformational analysis were truly reflecting receptor active states. This filtering process yielded 999, 1109, 1277, and 1057 ns of simulated time for subsequent analysis of WT:WT CXCR4, Csel2:Y7L, Cdyn:V3Y, and Csedy:V3Y:Y7L, respectively. PCA was performed on the cartesian coordinates of Cα and Cβ atoms of peptide ligands from receptor–peptide conformations selected by combining molecular dynamics trajectories from each of the studied systems. Representative models from the molecular dynamics trajectories were chosen as the highest density points in the space of principal components (PCs) 1 and 2. PCA was also performed individually for each of the systems studied with MD on the cartesian coordinates of Cα of peptide ligands. The PCA space was then clustered using a k-means clustering algorithm, with the optimal number of clusters being evaluated by the Calinski-Harabasz criterion. The variability explained by the first 2 PCs is shown in Supplementary Table 5. For each cluster, contact frequency between receptor and peptide

residues was calculated as the percent of frames for which a heteroatom of a given receptor residue is within 5 Å of a heteroatom of a given peptide residue.

## Mutual information calculation

We calculated mutual information (MI) from correlated motions extracted from MD simulations[38,70]. To calculate MI from torsional angles, a list of all backbone ($\varphi$ and $\psi$) and side chain ($X_1$ up to $X_5$ where applicable) torsion angles was built from the initial structure, the torsions were then extracted every 100 ps after removing the first 50 ns of every replica. The torsions were then histogrammed using 50 bins ($50 \times 50$ bins for two-dimensional histograms), and the marginal entropy was calculated using the following equation:

$$S_{\phi_i} = -R \sum_{n=1}^{B_i} P_{\phi_i}(n) \ln\left(\frac{P_{\phi_i}(n)}{h_\phi}\right) \qquad (5)$$

where $\phi_i$ is the torsion being sampled for residue $i$, $R$ is the gas constant, $B_i$ is the number of bins. $P_{\phi_i}(n)$ is the probability of finding $\phi_i$ in bin $n$ defined as: $P_{\phi_i}(n) = \frac{N_i(n)}{N}$, with $N_i(n)$ being the number of data-points/snapshots where $\phi_i$ falls in bin $n$ and $N$ the total number of datapoints/snapshots. $h_\phi$ is the width of each bin in the histogram defined by the torsion $\phi_i$. For two torsions $\phi_i$ and $\psi_j$ belonging to residues $i$ and $j$, the joint entropy is therefore defined as:

$$S_{\phi_i\psi_j} = -R \sum_{n=1}^{B_i} \sum_{m=1}^{B_j} P_{\phi_i\psi_j}(n,m) \ln\left(\frac{P_{\phi_i\psi_j}(n,m)}{h_\phi h_\psi}\right) \qquad (6)$$

where $P_{\phi_i\psi_j}(n,m)$ is the joint probability of finding $\phi_i$ in bin $n$ and $\psi_j$ in bin $m$. We then get the corresponding mutual information term $I_{\phi_i\psi_j}$:

$$I_{\phi_i\psi_j} = S_{\phi_i} + S_{\psi_j} - S_{\phi_i\psi_j} \qquad (7)$$

Correction for finite-size effects was also added:

$$\left\langle S^{observed} \right\rangle \approx S - \frac{M-1}{2N} \qquad (8)$$

where $\left\langle S^{observed} \right\rangle$ is the estimated entropy using $N$ datapoints and $M$ is the number of histogram bins with non-zero probability[71,72].

Another side effect of finite sample sizes is nonzero mutual information in independent datasets. To correct for this effect, we divide the observed MI space into 100 bins. In each MI bin, we randomly pick 5 dihedral pairs (or less if the bin has <5 samples) to represent the bin, and then for every dihedral pair $\phi_i$ and $\psi_j$, we shuffle the time series of one of the observed dihedrals and recalculate MI with the shuffled dihedral. This process is repeated until the shuffled dihedral MI converges, and then the average of the resulting MI over the chosen dihedral pairs approximates the nonzero independent MI for a given MI bin. This value of independent MI is then subtracted from all MI values belonging to the bin. These permutations can also be used as a test of significance for MI, and the percentage of MI values from the various permutations that are larger than the observed MI approximates a $p$-value[73]. We used an MI significance level of $p < 0.01$ in our analysis.

## Allosteric pathway and pipeline calculation

Allosteric pathways were calculated by first constructing a graph where nodes are residues (from either peptide or receptor) and where edges are formed between any pair of residues with significant MI that have their C$_\alpha$s within 10 Å. We then constructed pathways by minimizing the number of intermediate nodes while maximizing the sum of edge MIs between pairs of residues with significant MI whose C$_\alpha$s further than 10 Å apart using Dijkstra's shortest path algorithm[74].

Allosteric pathways were then sorted according to the MI of their terminal residues, and the number of pathways considered for further analysis in every system accounted for 85 % of cumulative mutual information. To cluster pathways into allosteric pipelines according to their closeness in the 3D structure, we define an overlap parameter, which is the percentage of nodes from two given pathways within a cutoff distance (7.5 Å). Overlapping allosteric pathways are clustered into allosteric communication pipelines using hierarchical clustering, and the strength of a pipeline is the number of pathways passing through it. We considered the top-ranking 20 pipelines for our analysis. Allosteric residues are defined as the residues with the largest number of allosteric pathways passing through them. Ligand and receptor residues are considered in contact if their heavy atoms are within 5 Å.

### Calculation of receptor cavity cross-sectional area
To quantify the conformational diversity of the ligand binding site between receptor variants, the ligand-binding pocket of the receptor models (from whose the ligand molecule was removed) was geometrically described using pyKVFinder[75]. Cavities were defined using a 1.2 Å probe over a cubic lattice (0.25 Å vertex). A 4.0 Å probe was used to define the entrance of the cavity.

### Expression constructs
WT CXCR4 with an N-terminal 3xHA-tag, $G\beta_3$-WT, and GNA15 subcloned into pcDNA3.1(+) were obtained from the cDNA Resource Center (Bloomsberg, PA). Designed CXCR4 variants and library point mutants were generated by site-directed mutagenesis. BRET fusion constructs for $G\alpha_{i1}$−91-Rluc8 and $G\gamma_9$-GFP2 were derived from optimized Tru-path constructs[76] and sub-cloned into pcDNA3.1(+) (Genscript Biotech).

### Modified peptides
Peptides were synthesized with C-terminal amidation (to reduce unwanted charge effects at the carboxy terminus) to generate wild-type and variants of the 17 N-terminal residues of CXCL12 (KPVSLSYRCPCRFFESH) (GenScript Biotech), a peptide known to elicit calcium mobilization and $G\alpha_i$ coupling signaling[20,21]. Lyophilized peptides were stored at −80 °C and resuspended in assay buffer on the day of the experiment.

### Enzyme-linked immunosorbent assay (ELISA) for receptor expression
Receptor expression was measured by ELISA in parallel for each experiment in a poly-D-lysine coated, white-walled, clear-bottom 96-well plate. Cells were fixed with 4% paraformaldehyde (EMS, ref: 15710) in PBS for 15 min at RT and blocked with 2% BSA for 45 min. After that, 45 min incubations, first with an anti-HA antibody (Thermofisher, ref: 26183) at a dilution of 1:500 followed by a second anti-mouse IgG antibody (CST, ref: 7076S) at 1:2000 dilution, were performed. Finally, chemiluminescence was recorded at the FlexStation3 after 10 min incubation with substrates A and B of SuperSignal West Pico PLUS kit (Thermofisher, ref: 34577). Average values from three replicates were normalized to WT HA3-CXCR4 expression.

### Calcium mobilization assays
40,000 HEK 293T cells (gift from Prof. Ted Wensel at Baylor College of Medicine) were transiently transfected with 50 ng WT or variant HA3-CXCR4, 10 ng GNA15 in pcDNA3.1(+). To equalize receptor surface expression, 75 ng of HA3-CXCR4 was used for the Csel2 variant. Cells were first seeded in 100 μL DMEM (Gibco, ref: 41965-039) 10 % FBS in a black-walled, clear-bottom 96-well plate coated with poly-lysine (Sigma, ref: P6407-5MG). Directly after cell loading, 50 μL of the mixture containing 0.5 μL Lipofectamine 2000 (Invitrogen, ref: 11668-019) and the DNA was added on top of the cells. The cells were incubated at

37 °C, 5% $CO_2$, and 95% relative humidity for 20 h, after which, media was refreshed with 150 μL DMEM + 10% FBS. Cells were assayed 48 h post-transfection. Cells were washed with 200 μL FLIPR6 buffer (HBSS + 20 mM HEPES, pH 7.4), then incubated at 37 °C, 5% $CO_2$, and 95% relative humidity for 2 h in 200 μL dye buffer according to the manufacturer's protocol (Molecular Devices, ref: R8190). Just before the assay, 5× concentrated peptide solutions were prepared in FLIPR6 buffer in a V-bottom 96-well plate. After incubation, peptide solutions were added at a rate of 16 μL/s after 30 s and fluorescence changes were monitored for 90 s after addition using microplate reader Flex-Station3 (Molecular Devices). The maximum response after correction by the mock-transfected condition was averaged from three replicates. Values were then plotted against selected concentrations and fitted to a sigmoidal curve using GraphPad Prism v9.

### Gαi dissociation BRET
40,000 HEK 293T cells were transiently transfected with WT or variant HA3-CXCR4, $G\alpha_{i1}$−91-Rluc8, $G\beta_3$-WT, and $G\gamma_9$-GFP2 in pcDNA3.1(+) at a ratio of 10:1:10:5 respectively. Cells were first seeded in 100 μL DMEM 10% FBS in a poly-lysine coated, white-walled, white-bottom 96-well plate. Directly after cell loading, 50 μL of the mixture containing Lipofectamine 2000 and the DNA was added on top of the cells. The cells were then left to incubate at 37 °C, 5% $CO_2$, and 95% relative humidity for 20 h, after which, 150 μL media was refreshed. Cells were assayed 48 h post-transfection. Cells were washed with 150 μL PBS, then 40 μL BRET buffer (HBSS, 0.2% Glucose) was added to each well. Coelenterazine 400a was first added at a final concentration of 2.5 μM and BRET ratios were measured once using Mithras[2] LB 943 plate reader. After the first measurement, 40 μL of a 3x concentrated agonist solution was added to each well, and BRET ratios were measured for another 30 min using a Mithras[2] LB 943 plate reader. Mock-transfected controls were subtracted from the data. Maximum values averaged from 3 replicates were then plotted against selected concentrations and fitted to a sigmoidal curve using GraphPad Prism v9.

### Full-length chemokine purification
CXCL12 and variants expressed in pMS211 (pET21a-based) construct and purified as previously described[23]. N-terminal His-tag and leader sequence cleaved with enterokinase to produce a final product with the correct N-terminal sequence. The final lyophilized protein was resuspended at 1 mg/ml in 0.1% BSA. Aliquots were snap frozen and stored at −80 °C.

### Peripheral blood mononuclear cells from healthy human donors
Buffy coats from de-identified healthy human volunteer blood donors were purchased from the Center of Interregional Blood Transfusion SRK Bern (Bern, Switzerland). Donors were de-identified and provided as a commercial entity, so no ethical approval is required.

### Generation of retroviral vectors
Retroviral constructs encoding HA3-CXCR4 wild-type or designed variants were generated using the In-Fusion HD Cloning Kit (Takara, ref: 638933). Sequences of interest from the expression constructs mentioned above were amplified by high-fidelity PCR (CloneAmp HiFi PCR Premix, ref: 639298). p-SFG retroviral backbone containing an IRES-ΔCD19 reporter gene was linearized by NotI-HF (NEB, ref: R3189S) and XhoI (NEB, ref: R0146S) restriction enzyme digestion (2–3 h at 37 °C). PCR fragments were gel purified from an agarose gel using the QIAquick Gel Extraction Kit (Qiagen, ref: 28706×4). Fragments of interest were assembled using the In-Fusion enzyme mix with the linearized backbone to generate the constructs of interest and transformed them into stellar competent cells. Plasmid DNA was purified from minipreps with QIAprep spin Miniprep Kit (Promega), and constructs were verified by sequencing (Microsynth).

## Generation of retroviral supernatant

Retroviral supernatant was produced by transient transfection of 293T cells as previously described[77]. In brief, 293T cells at 50% confluency were co-transfected with (1) the RDF plasmid encoding the RD114 envelope, (2) the Peg-Pam plasmid encoding MoMLV gag-pol, and (3) the SFG retroviral plasmid of interest (with LTRs and packaging signals), using GeneJuice transfection reagent (Merck, ref: 70967-3) according to manufacturer's instructions. Retroviral supernatants were harvested after 48 and 72 h of culture, filtered with 0.45 µM filter (Filtropur S, Sarstedt, ref: 83.1826), snap-frozen on a dry ice/100% ethanol mixture, and then stored at −80 °C until use, or used as fresh supernatant.

## Generation of human T cells expressing CXCR4 WT or variants

Peripheral blood mononuclear cells (PBMCs) were isolated from buffy coats by density gradient centrifugation (Lymphoprep, StemCell ref: 07851) and activated on plates coated with anti-CD3 (1 mg/ml, Biolegend, ref: 317347, clone: OKT3) and anti-CD28 (1 mg/ml, Biolegend, ref: 302934, clone: CD28.2) antibodies in T cell media (RPMI 10% FBS, 2 mM L-Glutamine, 1% Penicillin-Streptomycin) with IL-15 and IL-7 (Miltenyi Biotec, 10 ng/ml each, ref: 130-095-362 and ref: 130-095-765 respectively). The day before transduction, non-cell tissue culture treated 24-well plate (Greiner Bio-One, ref: 662102) was coated with retronectin (Takara Bio, ref: T100B) in PBS (7 µg/ml, 1 ml per well), and incubated overnight at 4 °C. Three days after activation, retronectin was removed and the plate was blocked with RPMI 10% FBS for 15 min at 37 °C. Then, media was removed and retroviral supernatant was centrifuged at 2000 × g, 1 h, 32 °C on retronectin-coated plates. Retroviral supernatant was gently removed and activated T cell suspension at 0.15 E$^6$ cells/ml was added, and centrifuged at 1000 × g, 10 min, 21 °C. Cells were incubated at 37 °C, 5% CO$_2$, and 95% relative humidity for 3 days. After 48–72 h of transduction, T cells were harvested and further expanded in T cell media containing IL-7 and IL-15. Transduced T cells were positively selected with a PE selection kit (EasySep, ref: 17684) and an anti-HA PE-conjugated antibody (Biolegend, ref: 901518, clone: 16B12) at a 1:50 dilution to enrich transduced T cells. Enrichment of surface expression in PE-selected cells was confirmed by flow cytometry using a BD FACS LSR II cytometer (BD Biosciences) and analyzed with FlowJo software (BD Biosciences).

## Migration assays

T cells transduced and selected for expression of CXCR4 variants from 3 to 6 donors were stained with 1 µM Vybrant DiO cell-labeling solution (Thermo, ref: V22886) in serum-free RPMI 1640 + GlutaMax. 40,000 cells in 75 µL were seeded in each well of 96-well Boyden chambers with 5.0 µm pores (Corning, ref: 3388)[44]. Reservoirs were filled with 200 µL serum-free RPMI 1640 + GlutaMax without chemoattractant or supplemented with 100 nM chemokine. The bottom of the attractant reservoir was imaged for migrated cells for 8 h with a Cytation 5 BioSpa (Biotek) at 37 °C with 5% CO$_2$. Fluorescent spots were counted over time and compared to the no-chemoattractant control to calculate the migration index (# migrated cells towards chemoattractant / # migrated cells in absence of chemoattractant). The peak migration indices averaged between 3 technical replicates for each transduced donor per chemoattractant concentration were plotted.

## Reporting summary

Further information on research design is available in the Nature Portfolio Reporting Summary linked to this article.

## Data availability

The authors declare that all data supporting the findings in this study are either presented within the article and its Supplementary Information files or available from the corresponding author on request. The following PDB entries were used for modeling: 4RWS [https://doi.org/10.2210/pdb4RWS/pdb], 4XT1 [https://doi.org/10.2210/pdb4XT1/pdb], 6LFO [https://doi.org/10.2210/pdb6LFO/pdb], 4UAI [https://doi.org/10.2210/pdb4UAI/pdb], and 7F1R [https://doi.org/10.2210/pdb7F1R/pdb]. Source data are provided with this paper.

## Code availability

The modeling, design and AlloDy softwares developed in this study together with a detailed Readme for running the simulations are available in the following GitHub repositories: https://github.com/barth-lab/CAPSens_design[78] and https://github.com/barth-lab/AlloDy[79].

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

## Acknowledgements
We thank the Barth lab members for helpful discussions. Funding: R.E.J. is supported by a Marie Skłodowska-Curie Individual Postdoctoral Fellowship (H2020 fund 588412). A.R. is supported by an EPFLglobaLeader doctoral fellowship. P.B. is supported by Swiss National Science Foundation grants (SNSF grants 31003A_182263 and 310030_208179), Swiss Cancer Research (KFS-4687-02-2019), funds from EPFL, and the Ludwig Institute for Cancer Research. C.A. receives funding from Swiss Cancer Research KFS-4542-08-2018-R, Stiftung für Krebsbekämpfung, Fondation Leenaards (Prix Leenaards 2021 pour la recherche médicale translationnelle), Helmut Horten Stiftung, Fondation Muschamp, ISREC Foundation, and the Department of oncology UNIL CHUV, University of Lausanne, Lausanne University Hospital.

## Author contributions
R.E.J. and P.B. designed the study; R.E.J. and A.F. developed and tested the method, R.E.J. and M.H. performed calculations; R.E.J., A.O., and A.R. performed the experimental validation of the designs; N.C. generated retrovirus and transgenic primary human T cells and analyzed results; C.A. supervised experiments with human T cells and analyzed results; R.E.J. and P.B. analyzed, interpreted results and wrote the manuscript.

## Competing interests
P.B. holds patents and provisional patent applications in the field of engineered T cell therapies and protein design. C.A. receives royalties from Immatics and holds patents and provisional patent applications in the field of engineered T cell therapies. All other authors declare no competing interests.
