## [Peer Review File · Nature Communications]

Computational design of dynamic receptor-peptide signaling complexes applied to chemotaxisREVIEWER COMMENTS

Reviewer #1 (Remarks to the Author):

The authors report a computational framework for designing conformationally dynamic biosensors, particularly GPRCs binding to peptidic ligands (CAPSens). They focus on the design of ultrasensitive CAPSens of chemotactic peptides for reprogramming cellular migration, in particular, on peptides for the CXCR4 receptor by basing the peptide designs on the N-terminal partially unstructured agonist region of the chemokine CXCL12. Overall, this is an exciting approach and, indeed, one solution to a challenging problem, i.e., the design of dynamic peptide-receptor complexes. It presents comprehensive experimental characterizations. I believe the work will be interesting for the community, but the details of the computational work should be re-arranged.

Besides, since the authors claim it is a computational protocol, it is somewhat contradictory that they do not release the code or package to use it in other scenarios or provide more details about the method in the main text. In its current form, I found it hard to find the computational details in the main text (beyond p.6 first paragraph and Fig. 1) and had to find most of them in the Methods section or later sections. The work is interesting, but it would benefit from reorganizing and emphasizing details on the computational protocol; seeing how robust it seems to elicit chemotaxis in human T cells, besides reorganizing the text, I besides suggest releasing the code to benefit the community.

Some other minor points that I found while reviewing this work, in line with the two comments above:

- p5. (i) I guess the authors started this work before the release of AlphaFold; otherwise, why not use AF or any other structure prediction method? Same for page 6, last paragraph.

- p.7 above. What does 'and design generation (1 and 2)' refer to?

- Fig. S1 is quite essential to understand the structural basis of this work, and it would help to bring it to the main text, perhaps by reorganizing Fig. 1.

- p.7 In 'Our design strategy by conformational selection focuses on the first shell of residues in contact with the peptide ligand, carries out independent combinatorial design calculations on single receptor-peptide conformations and identifies the few complexes with the strongest engineered binding interactions' it is unclear how the mutations in the first shell around the peptide happen. Do the authors

use Rosetta or any other physicochemical-based software? If this is the case a few sentences would clarify.

- p.8 'We rationally created and screened a computationally guided library of variants built from our initial ensemble of 9 receptor:peptide models. Each variant was designed by mutating a single predicted peptide binding and/or allosteric residue in the conformational ensemble and assayed for calcium mobilization' - I assume only first-shell residues are being mutated. How many mutants per position were considered? In short, how large the library was?

- p.8 in 'These results indicate that our approach can readily design highly sensitive sensors of the WT CXCL12 chemokine-derived peptide by optimizing both binding and signaling determinants.' it would help the reader to summarize the results of this section, which design was the most active of the three approaches (Csel, Cdyn, Csed) and then hypothesize why Csed showed more potency than Cdyn.

-p. 8 'From our computational models, we identified 2 sites P3 and P7 on the peptide scaffolds where novel and stronger contacts with the receptor binding pocket could be designed' unclear how these two spots are identified, or if they relate to previously mentioned residues (e.g. 113^{3.29}).

-p.9 'T cells transduced with the Csel2, Cdyn, and Csed designs displayed up to almost 4.7-fold increased migration towards 100 nM WT CXCL12. ' How many mutations to WT do these designs display, or what is their %identity to WT?

- p.11, it needs to be clarified a bit more how the Allody method works in the first paragraph of the section.

- p.27 The method uses a 'combined interface and peptide energy score.' How is this score computed? Also, p.28 first paragraph.

Reviewer #2 (Remarks to the Author):

This is an exceptional piece of work that greatly advances technologies aimed at understanding and repurposing allostery in protein ligand complexes. The example used by the authors (T cell engineering) makes the paper extremely impactful. I think the paper could be rewritten to improve reproducibility but also to communicate these very important ideas to a wider readership. I am recommending below

that the authors lead with their application and then discuss generalities later in the paper. I simply think this will help the authors reach a broader audience. See my comments below:

1. This paper is a significant advancement on so many levels (repurposing peptidergic receptors, dealing with very dynamic ligands and receptors missing inactive/active structure representations, and then applying all of this toward chemotaxis. However, it's a lot to unpack and I don't find it as readable (to the uninitiated) as it could be. I'd recommend a paper focused on repurposing a cytokine receptor to allow T cells to partition to a target tissue through chemotaxis. Start with the problem and then argue how chemokine receptors can be repurposed expressly for this purpose. Then later in the discussion section, describe the generality of the approach, perhaps in the context of peptidergic receptors. In this case, the authors would lead with some of the challenges associated with T cell therapy in cancer and provide a background to chemotaxis strategies. The authors write "..., efficient immune cell homing to and into cancers is one of the main bottlenecks in modern immunotherapy". If this were a focused paper on Immune cell homing, the authors could first write about chemotaxis and the challenges with T cell targeting, which they have currently relegated to references. Then there is the issue of the best strategy to take for chemotaxis in T cells. What kind of repertoire of chemokines exists in T cells and which one is prudent to repurpose in an engineered T cell? Would one take advantage of a natural gradient of chemokines that might exist in a tumor or would the plan be to also add the agonist to the tumor sites? The abstract as it stands describes the work as a way of repurposing a conformationally adaptable protein for a specific interaction and signal output. It's general but at the cost of being extremely vague. I'd rewrite it as generally applicable but lead with the opportunity of chemokine receptors and chemotaxis.

In what follows I have only minor questions which came up for me as I read the document (as a non protein-engineering reader).

2. Fig 1A delineates residues which modulate binding and residues which modulate activation. Presumably, more often than not, one would identify residues which perform both. This is later discussed in the paper but perhaps the authors would want to reflect this in the figure.

3. Fig 1B and Methods. This was more mysterious to me although it became clearer after reading everything. For example, the results in Figure 2 presumably do not refer to different CXCL12 peptides; rather they are simply obtaining ensembles of poses for the WT peptide as a function of mutations in the CXCR4 receptor?

4. Methods section

Constructing scaffolds

"the CXCR4 sequence was threaded onto the hybrid template structures..."

By threaded, do the authors mean to say that the peptide was sequentially docked along the protein binding interface? And what sort of side-chain/backbone flexibility was allowed during the process?

5. Docking

There are a huge number of degrees of freedom in a CXCL12 peptide with 11 disordered residues. While I understand that 3X3CL1 was used as template, I'm trying to understand how flexibility was handled during docking. I also don't understand the specific details of decoy generation. Presumably in the dock, the receptor is rigid?

It's also interesting that the authors discuss critical agonist contacts, known from structures. It would be tempting to implement these contacts as pseudo potentials and solve for peptide conformers much as is done by NMR from NOE constraints.

6. Loop rebuilding

"The number of bins at each sorting stage was adjusted such that 200-220 unique peptide conformations were selected for de novo loop modeling to build loop structures onto the initial scaffolds that best accommodate the bound peptide conformation (200 decoys per peptide-docked input). "

Here, the authors use position, shape, and orientation to bin their conformers. As these aren't one-dimensional criteria, I'm having trouble envisaging how this binning was performed.

I also can't connect this description to Figure 1B. Are the authors rejecting conformers after binning? "Loop rebuilding" implies to me refinement of the more dynamic regions.

7. Refinement of the Ensemble

"The cluster of models that best supported of observed changes in EC50 for each variant was used as an initial input for further flexible peptide docking refinement to identify optimal conformations for the WT, Csel2, library-selected and Csed designs"

This needs a discussion about how receptor sequences were explored alongside ensemble exploration. I would have guessed that the authors would try to get a sense of the WT peptide ensemble (~ 11-17 residues?), then identify hotspots, perhaps try something like Monte Carlo simulations to span conformational space, while allowing receptor/peptide flexibility, and then try mutating around hot spots. How do you know if a specific cluster of (structural) models supports the EC50? Do these results refer only to receptor mutations or later peptide super-agonist mutations?

8. "1000 decoys were generated and clustered and the lowest energy cluster was selected for further modeling" I can imagine many ways of generating decoys; how were the decoys generated? Presumably certain residues are conserved around all motifs?

9. page 4 "ensembles and predict" should read "ensembles and predicted"

10. Page 6 " As a proof of concept, we modeled and designed peptide ligand agonists starting from the N- terminal partially unstructured agonist region of the chemokine CXCL12 (Figure S1), which promotes strong activation of the CXCR4 receptor."

I gather from Figure S1 that the N-terminal 8 amino acids constitute the most buried end of CXCL12? Again some background to the CXCR4 receptor, cognate G proteins and arrestin pathways would be helpful.

11. Figure 2 and the associated discussion is very interesting. I think the figure/caption needs to be better annotated. Presumably WT refers to the receptor, not peptide. And I can't fathom the difference between Csel1 and Csel1 sites or Csel2 and Csel2 sites. Again, this figure refers entirely to mutations on the receptor, in the presence of WT peptide and if so, what length? I gather from Figure S2 that the trial peptides are all 17 residues?

12. Figure S2 x-axis is displayed as "mutation". Presumably the residues along the x-axis represent the CLCL12 1-16 or 2-17 WT residues? And the heatmap is a readout of specific contacts made with peptide residues where a normalized change in activity is observed?

13. Page 8 "cellular activity and bypass" should read "cellular activity while bypassing"?

14. Fig 3. Great results! The super-agonist mutations don't make sense to me with regard to Figure S2. For example, wouldn't V3 and Y7 mutations would be V4 and Y8 mutations, referring to Figure S2? Also, in Fig. 3E, both G β y and G α i stimulation point toward chemotaxis etc. Presumably there is some distinction between the two pathways that warrants comment?

15. Fig. 4. When the authors refer to P2, P3, P4, P7 etc I presume they are referring to residue numbers in the peptide? Is there not any way to even qualitatively describe MD (peptide) PCA, shown in Figure 4C?

16. C1, C2, and C3 refer to conformational substates, identified by PCA analysis of MD trajectories? Over what timescale? Is there some tangible way of understanding what these substates represent? For example, are there TM7, TM6, TM5 coordinates that define their topologies? Relatedly, in the AlloDy approach, what total timescale of MD simulations was used to establish pairwise correlations? The supplementary text suggests 300 ns which begs the question as to what C1, C2, and C3 substates might be.

Reviewer #3 (Remarks to the Author):

This manuscript by Jefferson et al involves using computational design approaches to design a dynamic peptide agonist-receptor pairing of mutant chemokine receptor CXCR4: CXCL12 that shows higher efficacy and potency to Gi secondary messengers and chemotaxis compared to the wild type CXCR4: CXCL12. The authors show that taking the dynamic ensemble into account for the mutant design generates a high efficacy and potent agonist: CXCR4 pairing compared to optimizing the binding affinity of the peptide alone. The significance of the study is in demonstrating the power of using a combination of extensive computational predictions followed by experimental testing that has led to proof of concept in engineering biosensors involving a flexible peptide and a GPCR involved in chemotaxis.

There are several concerns that need to be addressed in this work:

1. The biological significance of generating these sensors is not evident.

2. In the introduction the authors claim that they have built a biosensor that upon binding of a flexible peptide leads to selective, fine-tuned signaling. However, the outcome of the coupling of the mutant peptide to the mutant CXCR4 has not shown to be selective either to Gi among other G proteins and/or beta-arrestin.
3. The writing of the manuscript is pretty terse and highly technical and not understandable for a general reader of Nature Communications.
4. In figure 2A and B the authors claim that the CXCR4 residue position number 1.39 is an allosteric site, but it is really not allosteric since it is contacted by peptides in many other CC receptors.
5. At every stage of designing mutants there are several factors that are not clear: how many peptide:receptor combination mutations were designed and tested and how many predicted mutants turned out to be true positives?
6. On page 7, the last sentence, the authors state that "We rationally created and screened a computationally guided library of variants built from our initial ensemble of 9 receptor:peptide models. Where did these 9 receptor models come from? How were the residues to be mutated in the peptide and the receptor (both in the binding site and allosteric sites) selected? How were the mutants designed – by mutating the pairs of amino acids in the peptide and receptor? How were the mutants selected?"
7. On page 8, the statement that synthetic sensor-response systems would be useful for reprogramming cellular activity is a stretch since it is not known how these systems will trigger signaling pathways in the cellular system and more importantly in tissues.
8. There are many variables in modeling these systems - they were done as an ensemble model - how reliable are the shifts in peptide backbone and sidechain that the authors discuss in figure 4A?
9. How many structural conformations were used to calculate pocket cross section area for each mutant. Were MD simulations done on these mutants?
10. The statements "Our results support a view of receptor:peptide sensing where the inherent plasticity of the binding interface enables the efficient adaptation of contact networks in response to even limited changes in sequence space. Modeling and designing receptor:peptide interactions as conformationally dynamic complexes allows us to exploit this hallmark and readily evolve novel functional pairs." are exaggerated since limited scale MD simulations that the authors have performed cannot recapitulate the conformational changes from the agonist binding site to the G protein coupling site. Also, the authors have not discussed the changes in the intracellular regions of the receptor if any.
11. In page 11 in the paragraph starting with "We next assessed whether the engineered complexes.....", the authors need to discuss the data shown in Figure 5 - the inter-cluster RMSD is $\sim 14\text{\AA}$ and the 3.4\AA RMSD coming from the comparison of the residues in the binding site does not make sense. The reader is missing something here. Please explain.
12. The flexibility of the designed peptides shown in Fig. 5 – are they due to the starting structure being a modelled structure and not an experimental structure? How can the authors rule out this possibility.
13. AlloDy algorithm looks identical to prior work quoted in reference 58 and 59, yet there is no acknowledgement of the method in reference 58 in the main manuscript and in the Methods section.

14. The content of Figure 6 is not clear to a general reader of Nat Comm. it is very terse and complicated. Additionally, it is unclear what dynamism means? The conformational selection axis shown on top is very confusing as to how that figure is showing conformational selection. It may be easier if the authors show spread of the conformations in the PCA plot and show how the ligand selects a subset from the larger set in the same PCA plot.

15. On page 11, last sentence, the authors state that "Though the relative amounts of information passing through this conserved set of transmission hubs varies between designs, the allosteric" It is important to show data to substantiate these statements. How different are the allosteric strengths of the residues in the designed receptor:peptide mutant pairs?

16. The schematics shown in Fig. 1B are not clear and evident what they represent - for example, broken lines in fig.1B compared to solid lines showing allosteric communication. The legend should be expanded to explain the workflow.

17. In the legend for Figure 5, indicate to the reader what C1, C2 and C3 are. If the intercluster RMSD is about 14Å for the wild type CXC4:CXCL12 what is the 3.4Å number in RMSD coming from.

18. Figure 6 in the main manuscript and Figures S5 to S8 are complex to understand with no explanations provided in the legend for the color annotations in the figure

19. Not clear which structure was used for docking of peptide. Were all the structures from the lowest energy cluster used for docking? If yes, how many conformations were there in the lowest energy cluster and how different were they from one another.

20. Give BW numbering for residues throughout the manuscript including SI

21. In the Methods sections there are several queries listed below:

a. In the Loop building and ensemble relaxation to model mutual induced fit effects section the authors have used a scoring function that is called the "combined interface and peptide energy score" What is this function? Please expand.

b. In the same section there are statements such as "satisfaction of the experimentally informed constraints used in peptide docking" should be explained clearly. Which experimentally informed constraints were used? "what does satisfaction mean quantitatively in terms of tolerance used? The methods should be clear enough for other groups can use such procedures.

c. The sentences in the section titled "Computational design..." "Designable sites were identified on both the peptide and receptor sides of the different binding interfaces featured in the initial set of 9 CXCL12-bound receptor WT models. For each of these models, novel combination of amino-acids and conformations were searched concurrently for improving receptor:peptide association and signaling response. In silico mutagenesis was performed as previously described⁶⁴, allowing all possible residue substitutions at designable sites and selecting top scoring models for interface energy improvement from WT among 200 independent trajectories, such that scores converged for the top 10 % models." It is unclear where the initial set of 9 models came from? What does novel combination of amino acids and conformations mean? Which amino acid residues were chosen to be mutated in the peptide and in the receptor? Why were those chosen? When the authors say all possible residue substitutions, do they mean all the 19 other amino acids? Was the wild type residue also included in this list to treat the wild

type residue the same way as the mutants? What type of optimization was done after the mutation?
Where do the 200 independent trajectories come from?

d. Which programs were used for energy minimization and side chain packing.

e. Similarly, how were the mutation positions selected for Cdyn?

Overall, we are grateful to the reviewers for their insightful comments that we think we have fully addressed in this revised manuscript. Reviewers' text is in blue italic and our responses are in black and normal text.

REVIEWER COMMENTS

Reviewer #1 (Remarks to the Author):

The authors report a computational framework for designing conformationally dynamic biosensors, particularly GPRCs binding to peptidic ligands (CAPSens). They focus on the design of ultrasensitive CAPSens of chemotactic peptides for reprogramming cellular migration, in particular, on peptides for the CXCR4 receptor by basing the peptide designs on the N-terminal partially unstructured agonist region of the chemokine CXCL12. Overall, this is an exciting approach and, indeed, one solution to a challenging problem, i.e., the design of dynamic peptide-receptor complexes. It presents comprehensive experimental characterizations. I believe the work will be interesting for the community, but the details of the computational work should be re-arranged.

We thank the reviewer for the overall positive assessment of our study.

Besides, since the authors claim it is a computational protocol, it is somewhat contradictory that they do not release the code or package to use it in other scenarios or provide more details about the method in the main text. In its current form, I found it hard to find the computational details in the main text (beyond p.6 first paragraph and Fig. 1) and had to find most of them in the Methods section or later sections. The work is interesting, but it would benefit from reorganizing and emphasizing details on the computational protocol; seeing how robust it seems to elicit chemotaxis in human T cells, besides reorganizing the text, I besides suggest releasing the code to benefit the community.

Following the reviewer's suggestions, we have reorganized the main text to better emphasize and explain the computational methods (see pages 5-8). We also created a github repository for sharing the softwares and instructions to run the simulations. The link to the github repository is provided in the Code availability section.

Some other minor points that I found while reviewing this work, in line with the two comments above:

- p5. (i) I guess the authors started this work before the release of AlphaFold; otherwise, why not use AF or any other structure prediction method? Same for page 6, last paragraph.

Yes, this work was started before AlphaFold2 was published. To our knowledge, no other approach has been developed to specifically model and design flexible peptide-membrane receptor structures from homologs. A recent interesting study from the Furman lab indicates that AlphaFold can predict peptide:protein interactions despite not being trained on this task. Their results suggest that the main features of peptide binding can be implicitly captured as an extension of folding. However, our method is geared towards modeling flexible peptide interactions that do not involve strong patterns of unique static contacts such as those characteristic of folded polypeptide chains. Therefore, we believe that our approach provides a complementary approach to neural network-based methods trained on protein folded structures. We added a small paragraph on that topic in the discussion section on page 18.

- p.7 above. What does 'and design generation (1 and 2)' refer to?

The design generation refers to a specific round of iterative computational design. This is discussed in the subsequent paragraph, but we clarified that term when introduced.

- Fig. S1 is quite essential to understand the structural basis of this work, and it would help to bring it to the main text, perhaps by reorganizing Fig. 1.

We thank the reviewer for the suggestion. We inserted Fig S1 as caption A in Fig.1.

- p.7 In 'Our design strategy by conformational selection focuses on the first shell of residues in contact with the peptide ligand, carries out independent combinatorial design calculations on single receptor-peptide conformations and identifies the few complexes with the strongest engineered binding interactions' it is unclear how the mutations in the first shell around the peptide happen. Do the authors use Rosetta or any other physicochemical-based software? If this is the case a few sentences would clarify.

Mutations are indeed scored by Rosetta using the Rosetta Membrane scoring function. Those leading to decreased interface energies are selected by a Metropolis Monte Carlo Simulated Annealing protocol. We updated the main text accordingly on page 9 and added the following sentence: "Complexes are designed using Rosetta's Metropolis Monte Carlo Simulated Annealing protocol, scored using the RosettaMembrane energy function and selected if they led to decreased interface energies between the receptor and peptide".

- p.8 'We rationally created and screened a computationally guided library of variants built from our initial ensemble of 9 receptor:peptide models. Each variant was designed by mutating a single predicted peptide binding and/or allosteric residue in the conformational ensemble and assayed for calcium mobilization' - I assume only first-shell residues are being mutated. How many mutants per

position were considered? In short, how large the library was?

The results from all tested substitutions are summarized in Fig. S1. Receptor mutational sites were selected by proximity to any peptide conformation (i.e. residues with at least one heteroatom within 4 Å of the peptide) among the initial ensemble of models. The library comprised a total of 206 variants. We updated the main text on page 10 accordingly.

- p.8 in 'These results indicate that our approach can readily design highly sensitive sensors of the WT CXCL12 chemokine-derived peptide by optimizing both binding and signaling determinants.' it would help the reader to summarize the results of this section, which design was the most active of the three approaches (Csel, Cdyn, Csedy) and then hypothesize why Csedy showed more potency than Cdyn.

We added a sentence to summarize and discuss our findings at the end of the paragraph on page 10. The differences in potency do not correlate with the number of designed mutations since Csel2, Cdyn and Csedy incorporate 6, 3 and 7 mutations, respectively. Although it is not straightforward to identify which exact feature of the designs is responsible for the enhanced potency between Csel2 and Cdyn or Csedy, our results suggest that the conformational dynamism that we attempted to maintain at the binding interface was a key factor. This strategy favors multiple active conformations of the complex instead of stabilizing only a restricted subset through conformational selection that may not correspond to the most functionally active ones.

-p.8 'From our computational models, we identified 2 sites P3 and P7 on the peptide scaffolds where novel and stronger contacts with the receptor binding pocket could be designed' unclear how these two spots are identified, or if they relate to previously mentioned residues (e.g. 113^{3.29}).

Peptide mutations at positions P3 and P7 were selected because they were predicted to further increase binding interactions with Csel1 and Csel2. We updated the text accordingly on page 11.

-p.9 'T cells transduced with the Csel2, Cdyn, and Csedy designs displayed up to almost 4.7-fold increased migration towards 100 nM WT CXCL12.' How many mutations to WT do these designs display, or what is their %identity to WT?

The mutations of each design are summarized in Fig. 2A. Csel2, Cdyn and Csedy incorporate 6, 3 and 7 mutations, respectively. This information is now provided in the last summary paragraph at the end of page 10.

- p.11, it needs to be clarified a bit more how the Allody method works in the first paragraph of the section.

We added a short intro to clarify how Allody works in page 15 of the revised version.

- p.27 The method uses a 'combined interface and peptide energy score.' How is this score computed? Also, p.28 first paragraph.

The method uses Rosetta score terms “l_sc + pep_sc” from the FlexPepDock protocol. We updated the method on page 32 accordingly.

Reviewer #2 (Remarks to the Author):

This is an exceptional piece of work that greatly advances technologies aimed at understanding and repurposing allostery in protein ligand complexes. The example used by the authors (T cell engineering) makes the paper extremely impactful. I think the paper could be rewritten to improve reproducibility but also to communicate these very important ideas to a wider readership. I am recommending below that the authors lead with their application and then discuss generalities later in the paper. I simply think this will help the authors reach a broader audience. See my comments below:

1. This paper is a significant advancement on so many levels (repurposing peptidergic receptors, dealing with very dynamic ligands and receptors missing inactive/active structure representations, and then applying all of this toward chemotaxis. However, it's a lot to unpack and I don't find it as readable (to the uninitiated) as it could be. I'd recommend a paper focused on repurposing a cytokine receptor to allow T cells to partition to a target tissue through chemotaxis. Start with the problem and then argue how chemokine receptors can be repurposed expressly for this purpose. Then later in the discussion section, describe the generality of the approach, perhaps in the context of peptidergic receptors. In this case, the authors would lead with some of the challenges associated with T cell therapy in cancer and provide a background to chemotaxis strategies. The authors write "..., efficient immune cell homing to and into cancers is one of the main bottlenecks in modern immunotherapy". If this were a focused paper on Immune cell homing, the authors could first write about chemotaxis and the challenges with T cell targeting, which they have currently relegated to references. Then there is the issue of the best strategy to take for chemotaxis in T cells. What kind of repertoire of chemokines exists in T cells and which one is prudent to repurpose in an engineered T cell? Would one take advantage of a natural gradient of chemokines that might exist in a tumor or would the plan be to also add the agonist to the tumor sites? The abstract as it stands describes the work as a way of repurposing a conformationally adaptable protein for a specific interaction and signal output. It's general but at the cost of being extremely vague. I'd rewrite it as generally applicable but lead with the opportunity of chemokine receptors and chemotaxis.

We thank the reviewer for the overall positive assessment of our study and for the suggestion to refocus the manuscript towards one specific aspect of the findings. We agree that this would help the general readership to better grasp the novelty and potential impact of our work. Our study was initially

motivated by the lack of effective approach for modeling and designing flexible peptide-protein interactions at structurally-uncharacterized binding interfaces. The core of the study was dedicated to the development of the method. We then needed to validate the approach and selected the CXCR4-CXCL12 signaling axis because the binding complex remains structurally uncharacterized, and it performs important physiological functions. In the long term, we aim to apply the designed receptors developed in this study to enhance immune cell homing and cancer immunotherapies, but this is out of the scope of the present study as it would require extensive *in vivo* studies on multiple cancer models to fully assess the impact of cell migration on anti-tumor responses. Consequently, following the suggestions by the other reviewers, we decided to reorganize the paper as a broadly applicable method for receptor:peptide design and describe in more details the methodological aspects and novelty of the approach. We elaborated on the potential applications of engineered chemotactic biosensors in the discussion.

In what follows I have only minor questions which came up for me as I read the document (as a non protein-engineering reader).

2. Fig 1A delineates residues which modulate binding and residues which modulate activation. Presumably, more often than not, one would identify residues which perform both. This is later discussed in the paper but perhaps the authors would want to reflect this in the figure.

The activation residues in the binding pocket are in direct contact with the peptide and thus also binding residues. We updated the legend accordingly.

3. Fig 1B and Methods. This was more mysterious to me although it became clearer after reading everything. For example, the results in Figure 2 presumably do not refer to different CXCL12 peptides; rather they are simply obtaining ensembles of poses for the WT peptide as a function of mutations in the CXCR4 receptor?

This is correct. We clarified this in the figure 2 legend.

4. Methods section

Constructing scaffolds

"the CXCR4 sequence was threaded onto the hybrid template structures..."

By threaded, do the authors mean to say that the peptide was sequentially docked along the protein binding interface? And what sort of side-chain/backbone flexibility was allowed during the process?

As described in the subsequent peptide docking section, the peptide sequence of CXCL12 was threaded onto the CX3CL1 peptide structure from the US28-CX3CL1 active-state complex (PDB ID: 4XT1). This initial peptide model was then docked on a homology model of CXCR4 built from parts of

the US28 or CXCR4 structures. The peptide is docked from several starting positions covering the entire orthosteric binding site to avoid biasing towards a specific orientation. Flexible docking is carried out using the FlexPepDock protocol of Rosetta which involves extensive peptide rigid body and backbone motions, followed by side-chain rotameric motions of receptor and peptide and minimization over all conformational degrees of freedom at the binding interface. We clarified the main text and the method section accordingly on page 6 and pages 30-31, respectively.

5. Docking

There are a huge number of degrees of freedom in a CXCL12 peptide with 11 disordered residues. While I understand that 3X3CL1 was used as template, I'm trying to understand how flexibility was handled during docking. I also don't understand the specific details of decoy generation. Presumably in the dock, the receptor is rigid?

As mentioned above, the receptor backbone structure is also allowed to move during minimization, and side-chains are fully flexible at the receptor:peptide interface. While the starting CX3CL1 structure is used as a template for CXCL12, peptide internal motions around backbone and side-chain dihedrals are allowed during docking. This was clarified in pages 6 and 30-31.

It's also interesting that the authors discuss critical agonist contacts, known from structures. It would be tempting to implement these contacts as pseudo potentials and solve for peptide conformers much as is done by NMR from NOE constraints.

The constraints correspond to putative contacts inferred from mutagenesis studies (Wescott *et al.*, *PNAS*, 2016; Thiele *et al.*, *Br J Pharmacol*, 2014; Kufareva *et al.*, *PNAS*, 2014), but since we don't have high-resolution structural information on these contacts, we only applied a loose distance constraint potential. We now provide the list of constraints and constraint potential form on page 31-32.

6. Loop rebuilding

"The number of bins at each sorting stage was adjusted such that 200-220 unique peptide conformations were selected for de novo loop modeling to build loop structures onto the initial scaffolds that best accommodate the bound peptide conformation (200 decoys per peptide-docked input)."

Here, the authors use position, shape, and orientation to bin their conformers. As these aren't one-dimensional criteria, I'm having trouble envisaging how this binning was performed.

I also can't connect this description to Figure 1B. Are the authors rejecting conformers after binning?

"Loop rebuilding" implies to me refinement of the more dynamic regions.

We wrote a peptide conformation selection section on pages 32-33 which describes the 3 metrics used for binning peptide conformations. In summary, this step enables identification of peptide conformations that occupy diverse positions in the binding pocket. Through that process, we diversify the peptide conformations that are selected and we avoid biasing the selection of peptide poses that occupy similar positions in the pocket or adopt similar structures. Only the decoy with the best combined interface and peptide energy score is selected from each bin, and the other ones are discarded. Overall, this stage selects 200-220 decoys from a total of 10,000 generated structures. The loop rebuilding stage (page 33) is performed from each of the selected peptide-bound receptor scaffold structures.

7. Refinement of the Ensemble

"The cluster of models that best supported of observed changes in EC₅₀ for each variant was used as an initial input for further flexible peptide docking refinement to identify optimal conformations for the WT, Csel2, library-selected and CsedY designs"

This needs a discussion about how receptor sequences were explored alongside ensemble exploration. I would have guessed that the authors would try to get a sense of the WT peptide ensemble (~ 11-17 residues?), then identify hotspots, perhaps try something like Monte Carlo simulations to span conformational space, while allowing receptor/peptide flexibility, and then try mutating around hot spots. How do you know if a specific cluster of (structural) models supports the EC₅₀? Do these results refer only to receptor mutations or later peptide super-agonist mutations?

We hope that our new detailed description of the modeling & design pipeline in the main text will clarify how the peptide bound receptor ensemble is selected and then used as a starting point for the 2 design strategies.

Concerning the model selection based on EC₅₀, because all mutations were designed at positions that directly contact the peptide, we hypothesized that changes in ligand EC₅₀ should be dominated by changes in ligand binding affinity. In our simulations, we predicted binding energy changes from WT which we therefore compared to changes in EC₅₀. Since the designed mutations triggered large increases in potency, we selected the models that displayed substantial decrease in binding energies (i.e. from -3.9 Rosetta Energy Units for Csel2 to -6 Rosetta Energy Units for Cdyn). This refinement was carried out prior to the design of super-agonist mutations.

8. "1000 decoys were generated and clustered and the lowest energy cluster was selected for further modeling" I can imagine many ways of generating decoys; how were the decoys generated?

Presumably certain residues are conserved around all motifs?

Each decoy represents the outcome of an independent trajectory, i.e. an independent attempt at loop building (in the quoted case) using the loop modeling protocol of the software Rosetta. That protocol

uses a Metropolis Monte Carlo algorithm and requires multiple independent trajectories to assess convergence to low energy structures. Here the goal of this step was to rebuild missing or truncated intracellular loops from the hybridized scaffold. No constraints were made on the conformation of the ICLs remodeled and the sequences were fixed to that of the WT CXCR4.

9. page 4 "ensembles and predict" should read "ensembles and predicted"

This has been fixed.

10. Page 6 " As a proof of concept, we modeled and designed peptide ligand agonists starting from the N- terminal partially unstructured agonist region of the chemokine CXCL12 (Figure S1), which promotes strong activation of the CXCR4 receptor."

I gather from Figure S1 that the N-terminal 8 amino acids constitute the most buried end of CXCL12? Again some background to the CXCR4 receptor, cognate G proteins and arrestin pathways would be helpful.

We thank the reviewer for this suggestion. We now added Fig S1 as a caption to Figure 1 where we highlight the position of the first 8 residues that are the most buried in the orthosteric binding pocket. We also added some basic information on CXCR4 signaling in the main text on page 8.

11. Figure 2 and the associated discussion is very interesting. I think the figure/caption needs to be better annotated. Presumably WT refers to the receptor, not peptide. And I can't fathom the difference between Csel1 and Csel1 sites or Csel2 and Csel2 sites. Again, this figure refers entirely to mutations on the receptor, in the presence of WT peptide and if so, what length? I gather from Figure S2 that the trial peptides are all 17 residues?

The peptide length is noted in the X-axis of the dose response curves but we now also mention the peptide length in the Figure legend. We renamed CselX sites to "CselX tested mutations" to avoid any confusion.

12. Figure S2 x-axis is displayed as "mutation". Presumably the residues along the x-axis represent the CLCL12 1-16 or 2-17 WT residues? And the heatmap is a readout of specific contacts made with peptide residues where a normalized change in activity is observed?

The heatmap summarizes the measured activities of receptor mutants upon sensing the WT peptide. The x-axis is the substituted amino-acid type for a particular receptor residue mentioned on the y axis.

13. Page 8 "cellular activity and bypass" should read "cellular activity while bypassing"?

This has been fixed.

14. Fig 3. Great results! The super-agonist mutations don't make sense to me with regard to Figure S2. For example, wouldn't V3 and Y7 mutations would be V4 and Y8 mutations, referring to Figure S2? Also, in Fig. 3E, both Gβγ and Gai stimulation point toward chemotaxis etc. Presumably there is some distinction between the two pathways that warrants comment?

As we clarified in #12, Fig S2 refers to receptor and not peptide variants. We carried out Gai activation and Ca²⁺ release triggered by the release of Gβγ from the Gi heterotrimer because they provide complementary measurements of G-protein activation. This was clarified in the main text on page 8.

15. Fig. 4. When the authors refer to P2, P3, P4, P7 etc I presume they are referring to residue numbers in the peptide? Is there not any way to even qualitatively describe MD (peptide) PCA, shown in Figure 4C?

Yes. PX refer to residue numbers in the peptide. We updated the legend accordingly. We understand the reviewer's point but we found it hard to qualitatively describe a PCA metric. We think that the best structural view conveying these differences are the diverse peptide conformations shown in Fig.4.

16. C1, C2, and C3 refer to conformational substates, identified by PCA analysis of MD trajectories? Over what timescale? Is there some tangible way of understanding what these substates represent? For example, are there TM7, TM6, TM5 coordinates that define their topologies? Relatedly, in the AlloDy approach, what total timescale of MD simulations was used to establish pairwise correlations? The supplementary text suggests 300 ns which begs the question as to what C1, C2, and C3 substates might be.

The conformational substates refer to ligand conformations only (the most dynamic part of the simulation) and are identified by PCA analysis of the MD trajectories. The simulations correspond to at least 1500 ns cumulated trajectory, which are also used by AlloDy to predict pathways. These timescales are similar to those reported for allosteric pathway calculations based on Mutual Information carried out on other GPCRs (Bhattacharya & Vaidehi, *Biophysical J.* 2014; Chen *et al.*, *Nat Chem Biol* 2020). To stringently assess whether these timescales would be sufficient to calculate Mutual Information, we performed a convergence analysis of the first order and second order entropies for each substate (**Figure S4**). We observed less than 10% variability from the final entropy value over the last 500 frames (50 ns), which is a strong indication of convergence and suggest that our MI and subsequent pathway calculations are reliable.

Importantly, during these simulations, the receptor does not undergo substantial conformational changes and remains in the active state as we now show in **Figure S3** through the measurement of canonical TM6 and TM7 distance signatures of receptor activation. Consequently, our analysis identifies ensembles of conformations that are adopted by the peptide in the active state and enables

a qualitative comparison of the levels of conformational heterogeneity or dynamics in the active state between variants. We clarified this analysis and its implication in the main text on page 13.

Reviewer #3 (Remarks to the Author):

This manuscript by Jefferson et al involves using computational design approaches to design a dynamic peptide agonist-receptor pairing of mutant chemokine receptor CXCR4: CXCL12 that shows higher efficacy and potency to Gi secondary messengers and chemotaxis compared to the wild type CXCR4: CXCL12. The authors show that taking the dynamic ensemble into account for the mutant design generates a high efficacy and potent agonist: CXCR4 pairing compared to optimizing the binding affinity of the peptide alone. The significance of the study is in demonstrating the power of using a combination of extensive computational predictions followed by experimental testing that has led to proof of concept in engineering biosensors involving a flexible peptide and a GPCR involved in chemotaxis.

There are several concerns that need to be addressed in this work:

1. The biological significance of generating these sensors is not evident.

We expanded on the biological significance of the biosensors in the discussion section on page 19. Briefly, poor immune cell homing to tumor sites has been recognized as a major limitation of current T cell therapies. In the long run, we aim at engineering immune cells with these ultra-sensitive chemokine-sensing receptors for enhanced tumor homing and anti-tumor responses.

2. In the introduction the authors claim that they have built a biosensor that upon binding of a flexible peptide leads to selective, fine-tuned signaling. However, the outcome of the coupling of the mutant peptide to the mutant CXCR4 has not shown to be selective either to Gi among other G proteins and/or beta-arrestin.

We agree with the reviewer that the wording may be confusing as we did not intend to imply that we had designed pathway-selective biosensors. We removed the word selective and reworded the sentence as follows in page 4: "In the long run, we aim to design custom-built modular biosensors that can link binding of a flexible peptide input signal to fine-tuned and complex cellular responses through genetically encoded single-receptor domains." We also updated the last introduction paragraph to clarify its meaning.

3. The writing of the manuscript is pretty terse and highly technical and not understandable for a general reader of Nature Communications.

We understand the reviewer's point. To address this concern, we rewrote part of the text, expanding on the approach and providing a higher-level description of the methods. We also added conclusions to each section summarizing each finding and further developed the discussion.

4. In figure 2A and B the authors claim that the CXCR4 residue position number 1.39 is an allosteric site, but it is really not allosteric since it is contacted by peptides in many other CC receptors.

We removed any reference to position 1.39 to avoid any confusion.

5. At every stage of designing mutants there are several factors that are not clear: how many peptide:receptor combination mutations were designed and tested and how many predicted mutants turned out to be true positives?

The tested mutations are listed in Fig. 2A but we clarified in the main text the design process involving the 2 strategies and provided the number of tested variants and success rate. Specifically, we added the following sentences: Page 9: "Overall, we tested a total of 19 designs from this conformational selection approach and reached a success rate of 37%". Page 10: "Overall, we tested a total of 15 designs from this conformational dynamism approach and reached a success rate of 33%."

6. On page 7, the last sentence, the authors state that "We rationally created and screened a computationally guided library of variants built from our initial ensemble of 9 receptor:peptide models. Where did these 9 receptor models come from? How were the residues to be mutated in the peptide and the receptor (both in the binding site and allosteric sites) selected? How were the mutants designed – by mutating the pairs of amino acids in the peptide and receptor? How were the mutants selected?"

To address the reviewer's point, we provide a more detailed high-level description of the approach in the main text and additional details in the method (see page 34). The 9 models correspond to the top-scoring of the most populated clusters of models (>30 members) that satisfied experimentally derived constraints (<10 REU penalty) and made at least one key contact to the 2 N-terminal peptide residues (known to be essential for activity). Residues to be mutated were selected if they were in direct contact with either the peptide or the receptor. A direct contact is defined if 2 heteroatoms are within 4 Angstrom from each other. Two design approaches were pursued: in our first approach, mutations on the receptor were designed to enhance the interactions with the native peptide. In our second approach, both receptor and peptide were mutated to achieve stronger binding complexes. The mutants were selected if they led to enhanced calculated interaction energies and did not destabilize the structure of either the peptide or the receptor.

7. On page 8, the statement that synthetic sensor-response systems would be useful for reprogramming cellular activity is a stretch since it is not known how these systems will trigger signaling pathways in the cellular system and more importantly in tissues.

We do not fully understand the reviewer's comments. It seems to us that the ability to potentiate cell migration is a clear indication that our designed biosensors can modulate cellular activity.

Nevertheless, to be more specific in that paragraph on page 11, we modified the corresponding sentence to: "Such synthetic sensor-response systems would provide orthogonal solutions for modulating signal transductions ...".

8. There are many variables in modeling these systems - they were done as an ensemble model - how reliable are the shifts in peptide backbone and sidechain that the authors discuss in figure 4A?

To enable the structural comparison highlighted in Fig.4, we carried out up to 1.9 microsecond MD simulations of the complexes and performed a PCA analysis of the trajectories. The 3 most populated clusters of peptide conformations are shown in Fig.4. The backbone and side-chain conformations of representative peptide poses from each of these clusters are overlaid and shown next to the PC plots. Thus, the shifts in peptide conformations that we describe in Fig.4 are representative of the most populated conformations of the peptide during the simulations and do not correspond to outliers.

9. How many structural conformations were used to calculate pocket cross section area for each mutant. Were MD simulations done on these mutants?

The pocket cross-sectional area calculations in Fig.5 were performed using the same models described in Fig. 4 and extracted by PCA analysis of the MD simulations. We clarified the text and figure legends accordingly.

10. The statements "Our results support a view of receptor:peptide sensing where the inherent plasticity of the binding interface enables the efficient adaptation of contact networks in response to even limited changes in sequence space. Modeling and designing receptor:peptide interactions as conformationally dynamic complexes allows us to exploit this hallmark and readily evolve novel functional pairs." are exaggerated since limited scale MD simulations that the authors have performed cannot recapitulate the conformational changes from the agonist binding site to the G protein coupling site. Also, the authors have not discussed the changes in the intracellular regions of the receptor if any.

First, we would like to clarify how we performed the MD simulations. We started the simulations from peptide-receptor complex structures modeled in the active state. As shown in Fig. S3 through the measurement of canonical TM6 and TM7 distance signatures of receptor activation in the intracellular region, the receptor structure remained in the active state during the simulations. We did not attempt

to model conformational transitions between functional states that cannot be captured within the timescale of our simulations as correctly stated by the reviewer. The level of conformational diversity that we observed mostly came from the extracellular side where the peptide adopted bound conformations that are sensitive to the designed binding interactions in the binding pocket. We clarified these points in the main text on page 13. Through this sentence, we were merely suggesting that the conformational flexibility of the receptor-peptide binding interface may facilitate the creation of novel binding contacts through sequence changes during computational design. In fact, performing the same design calculations on more conformationally restrained binding interfaces would typically not give rise to such levels of structural adaptation. Nevertheless, to avoid any confusion or misinterpretation, we toned down the conclusion of this paragraph, which now reads: "Overall, these findings suggest that the high conformational plasticity of the CXCR4: CXCL12 binding interface may facilitate the adaptation of contact networks in response to even limited changes in receptor and peptide sequence space through mutual induced fit."

11. In page 11 in the paragraph starting with "We next assessed whether the engineered complexes.....", the authors need to discuss the data shown in Figure 5 - the inter-cluster RMSD is ~14Å and the 3.4Å RMSD coming from the comparison of the residues in the binding site does not make sense. The reader is missing something here. Please explain.

We thank the reviewer for pointing out this apparent discrepancy. The 3.4 Å RMSD was referring to the peptide conformational diversity within each cluster. We should report instead the inter-cluster RMSD. We modified the text accordingly.

12. The flexibility of the designed peptides shown in Fig. 5 – are they due to the starting structure being a modelled structure and not an experimental structure? How can the authors rule out this possibility.

The models used to characterize peptide conformational flexibility in Figures 4 to 6 are strictly coming from equilibrated MD trajectories so we expect that any slight inaccuracies from the initial Rosetta design models would fade out rapidly at the beginning of the simulation. As described in the methods, the first 50 ns of the simulations were discarded from PCA calculations and path analysis. To directly assess whether the peptide conformational diversity would be the consequence of modeled structures, we performed similar equilibrium MD simulations starting from the experimental structure of the CCR5 bound to the N terminal tail of the chemokine RANTES to mimic our CXCR4-CXCL12 tail complex. During the 1500 ns of the simulations, RANTES explores a relatively large conformational space. PCA analysis of the peptide conformational ensembles identified 3 major clusters of conformations separated by up to 12.4 Å (Fig. S5), which is similar to the structural diversity observed for the WT CXCL12. Therefore, these results suggest that the high CXCL12 peptide flexibility in our simulations

may not be the result of our starting modeled structures but reflect intrinsic structural properties of the molecules. Nevertheless, since we cannot completely rule out that some features of the observed conformations may still be biased by inaccuracies in the initial predicted models, we added a sentence in the discussion section on page 18 reminding the readership that these observations are coming from predicted structures and are subjected to potential inaccuracies.

13. AlloDy algorithm looks identical to prior work quoted in reference 58 and 59, yet there is no acknowledgement of the method in reference 58 in the main manuscript and in the Methods section.

We had cited references 58 and 59 in the methods section on page 38 where we described the Mutual information calculation. We now also cite that work (now ref 38) when introducing AlloDy in the main text on page 15. However, while AlloDy is indeed based on similar concepts than the method described in ref 38, it comprises a number of additional features that should enhance the stringency and reproducibility of the path analysis. For example, AlloDy includes correction for finite-size effects, statistical tests for mutual information significance and criteria ensuring convergence of the 1st and 2nd order entropies calculations. Lastly, AlloDy carries out automatically the ligand PCA and the GPCR activation state analysis. These specific features in AlloDy ensure also a fair comparison between variants.

14. The content of Figure 6 is not clear to a general reader of Nat Comm. it is very terse and complicated. Additionally, it is unclear what dynamism means? The conformational selection axis shown on top is very confusing as to how that figure is showing conformational selection. It may be easier if the authors show spread of the conformations in the PCA plot and show how the ligand selects a subset from the larger set in the same PCA plot.

We agree with the reviewer that Figure 6 was complex and attempting to convey too many messages. We now split the results shown in former Figure 6 into Figures 4 and 6. Specifically, the results on conformational flexibility at the peptide binding pocket and their interpretation in terms of contact networks are now discussed in pages 13-14 and displayed in Figure 4. The prediction of allosteric pathways running through the receptor for the different designed complexes are discussed in pages 15-16 and displayed more schematically in Figure 6. Our definition of dynamism refers to the high level of conformational flexibility and heterogeneity obtained with our second design strategy by contrast to the conformational selection obtained by our first design approach. To avoid any confusion, we now display these axes in Figure 4 next to the PCA plots where the peptide conformational diversity is analyzed and discussed.

15. On page 11, last sentence, the authors state that "Though the relative amounts of information passing through this conserved set of transmission hubs varies between designs, the allosteric" It

is important to show data to substantiate these statements. How different are the allosteric strengths of the residues in the designed receptor:peptide mutant pairs?

Through this sentence, we are merely commenting on the fact that despite different hubcores, the topology of the main paths remained very similar between variants. While our pathway comparison remains very qualitative, we now also report these hubcores in Table S2.

16. The schematics shown in Fig. 1B are not clear and evident what they represent - for example, broken lines in fig.1B compared to solid lines showing allosteric communication. The legend should be expanded to explain the workflow.

We modified Figure 1 to provide a schematic view of the design strategies in panels C, D, E. We also removed the broken lines to simplify the depiction of allosteric pathways in panel F. We updated the figure legend to describe in more detail each panel. Concerning the method workflow (now in panel F), we now provide a detailed explanation of each step in the main text because that level of details would not fit in a figure legend format.

17. In the legend for Figure 5, indicate to the reader what C1, C2 and C3 are. If the intercluster RMSD is about 14Å for the wild type CXC4: CXCL12 what is the 3.4Å number in RMSD coming from.

We updated the legend accordingly.

18. Figure 6 in the main manuscript and Figures S5 to S8 are complex to understand with no explanations provided in the legend for the color annotations in the figure

As described in point #14, we simplified Figure 6 and updated the legend of these figures accordingly.

19. Not clear which structure was used for docking of peptide. Were all the structures from the lowest energy cluster used for docking? If yes, how many conformations were there in the lowest energy cluster and how different were they from one another.

As we described in the method section, an initial template structure built by homology to existing active state receptor structures is used for docking peptides. Subsequent peptide docking generates an ensemble of receptor-peptide poses that are clustered and “diversified” to retain high conformational diversity of the peptide position prior to loop rebuilding and complex relaxation. The loop modeling approach ensures that loops are rebuilt in the presence of a large diversity of possible peptide poses to capture conformational selection effects and then induced fit effects when the receptor structure is allowed to relax in presence of the peptide. After loop rebuilding and complex relaxation, all models for each combination of receptor scaffold and experimentally derived constraint set (i.e. typically 20,000) are filtered by energy and the lowest 10% are clustered. We obtained 9 major clusters that differed between 0.65 and 2.55 Å RMSD.

20. Give BW numbering for residues throughout the manuscript including SI

We now provide BW notations throughout the manuscript.

21. In the Methods sections there are several queries listed below:

a. In the Loop building and ensemble relaxation to model mutual induced fit effects section the authors have used a scoring function that is called the “combined interface and peptide energy score” What is this function? Please expand.

We specifically used the ‘l_sc + pep_sc’ score terms from the Rosetta FlexPepDock protocol scoring function. We updated the method section accordingly page 32.

b. In the same section there are statements such as "satisfaction of the experimentally informed constraints used in peptide docking" should be explained clearly. Which experimentally informed constraints were used? "what does satisfaction mean quantitatively in terms of tolerance used? The methods should be clear enough for other groups can use such procedures.

We now provide in the method section on pages 31-32 the list of constraints, the constraint mode of Rosetta and corresponding equation applied during modeling. Constraints were considered satisfied if the remaining distance violations in the final models would not lead to constraint scores higher than 10 Rosetta Energy Units.

c. The sentences in the section titled “Computational design...”, “Designable sites were identified on both the peptide and receptor sides of the different binding interfaces featured in the initial set of 9 CXCL12-bound receptor WT models. For each of these models, novel combination of amino-acids and conformations were searched concurrently for improving receptor:peptide association and signaling response. In silico mutagenesis was performed as previously described⁶⁴, allowing all possible residue substitutions at designable sites and selecting top scoring models for interface energy improvement from WT among 200 independent trajectories, such that scores converged for the top 10 % models.” It is unclear where the initial set of 9 models came from?

We addressed this question in point #6.

What does novel combination of amino acids and conformations mean?

As for most computational design approaches based on Rosetta, search in sequence and conformational spaces are performed concurrently using a Metropolis Monte Carlo Simulated Annealing protocol. Combinations of amino-acids and conformations (represented as discrete rotamers) at sites specified by the user are randomly selected by the algorithm followed by gradient-

based energy minimization over all conformational degrees of freedom. The energy of the resulting structure is used to decide whether the move should be accepted.

Which amino acid residues were chosen to be mutated in the peptide and in the receptor? Why were those chosen? When the authors say all possible residue substitutions, do they mean all the 19 other amino acids? Was the wild type residue also included in this list to treat the wild type residue the same way as the mutants? What type of optimization was done after the mutation? Where do the 200 independent trajectories come from?

Any mutation is systematically compared to the WT. To ensure the sequence search is unbiased, all 20 residues were allowed at designable sites including the native one, so the method can decide to keep it. Since the search in sequence and structure space is stochastic, 200 independent trajectories are typically performed to ensure convergence of the Monte Carlo search. We updated the method section accordingly.

d. Which programs were used for energy minimization and side chain packing.

The repacking and minimization protocols of Rosetta are used throughout the study. We now cite Rosetta and RosettaMembrane in the method.

e. Similarly, how were the mutation positions selected for Cdyn?

We selected all the receptor positions directly contacting (i.e. within 4 Å heteroatom distance) the different peptide binding poses obtained from the initial 9 clusters of models. We updated the method section accordingly on page .

REVIEWERS' COMMENTS

Reviewer #1 (Remarks to the Author):

The authors have considerably improved the manuscript, and I believe it will be of interest to a broad readership. Thanks for making your code available to the community.

Reviewer #2 (Remarks to the Author):

I have no further comments. All of my previous questions/comments have been satisfactorily answered and the revised version is considerably improved in clarity.

Reviewer #3 (Remarks to the Author):

The authors have revised the manuscript extensively to make it readable for broad readership. The methods are clearly delineated. The authors have addressed all my suggestions and concerns.